# IntuitiveGraphLLM: Intuitive Graph-based Text Representation with Large Language Model

## Abstract

Graphical representations of text can sharpen the inductive biases of large language models (LLMs), yet most graph-based approaches rely on co-occurrence, order, or position alone and therefore over-connect unrelated tokens while missing conceptually salient links. We introduce Intuitive Graphs (IGs)—graphs that explicitly encode both ($i$) structural context (local order/proximity/position) and ($ii$) conceptual relevance (semantic affinity in embedding space)—and IntuitiveGraphLLM, a framework that builds, encodes, and fuses IGs with pretrained LLMs. Given a passage, we first construct IGs by pruning structure-induced edges with a semantic gate based on cosine similarity between token (or span) embeddings, yielding sparse, human-plausible graphs. We then obtain initial node features from contextual embeddings and apply Graph Attention Networks (GATs) to emphasize informative nodes/edges to produce graph-level features. Finally, we perform hybrid fusion by integrating graph-level embeddings with LLM-based contextual representations, enabling the model to leverage complementary structural and conceptual signals. We evaluate our approach on five benchmark datasets spanning short and long documents and class-imbalance settings. Across benchmarks, IntuitiveGraphLLM consistently improves over strong text-only and graph-only baselines; gains persist under varied IG constructions, node embeddings, GAT depths/heads, and LLM backbones, with ablations confirming that IG is the key driver of performance and reduced edge noise. IntuitiveGraphLLM provides a principled, interpretable way to make text graphs both contextual and conceptually grounded, translating into more faithful reasoning and stronger downstream accuracy.

## 1 Introduction

In recent year, the advancement of Large Language Models (LLMs) has led to significant performance improvments across a wide range of NLP tasks and domains, including sentiment analysis Cai et al. (2024); Rahman et al. (2024), biomedical retrieval Xu et al. (2024), question answering Robinson & Wingate (2023), code comprehension Du et al. (2024), summarization and generation Tu et al. (2024); He et al. (2024), and translation and text synthesis Papi et al. (2024). The scaling data and model size have further expanded their capabilities Wei et al. (2022b); Bubeck et al. (2023). Consequently, LLMs have attracted widespread attention from both academia Wei et al. (2022a); Zhao et al. (2023) and industry Achiam et al. (2023).

To adapt general-purpose LLMs for downstream tasks, numerous approaches have emerged. Beyond full-parameter fine-tuning, prompt- and prefix-based methods steer frozen models through learned prompts Lester et al. (2021); Li & Liang (2021). Few-shot approaches enable an LLM to become a domain-specific model with limited examples Brown et al. (2020). Parameter-efficient techniques that freeze pretrained weights and learn small rank-decomposition adapters reduce training cost while preserving performance Tian et al. (2024). Zhu et al. (2024); He et al. (2025) introduce novel parameter- and memory-efficient methods, such as ENGINE and UniGraph, which integrate LLMs with GNNs for textual graphs. Complementary directions integrate structured signals—e.g., knowledge graphs, hybrid feature pipelines, recurrent layers, and layer-specific adjustments—to enhance the structural and functional capacity of LLMs Bugueño & de Melo (2023); Rahman et al. (2024). Despite the countless successes, LLMs often fail to retrieve and reason about the actual and complex semantics (or relationships) expressed in the text Lewis et al. (2020); Pan et al. (2024). Graph-based representations offer a natural way to externalize and manipulate relational structure Ji

et al. (2021), but conventional text graphs, built from co-occurrence windows, sequential adjacency, or positional heuristics, can over-connect unrelated tokens and propagate noise. For example, in *"The hike was exhausting, but the view at the top was breathtaking,"* the conjunction *but* introduces a discourse contrast; capturing such structure-aware, conceptually meaningful relations is essential for faithful interpretation Bugueño & de Melo (2023). Structure-aware graph encodings used in retrieval-augmented generation Baek et al. (2023); Lewis et al. (2020) may include extraneous noise, degrading LLM performance Tian et al. (2024).

We propose IntuitiveGraphLLM, a framework that helps LLMs extract useful knowledge from Intuitive Graph (IG) representations of text. IGs first construct structure-driven graphs (e.g., windowed, sequential, position) and then apply a semantic gate, a cosine-similarity filter in embedding space, to retain edges that are both contextual and conceptually relevant. We initialize node features with domain-specific and contextual embeddings, and process IGs using Graph Attention Networks (GATs) Veličković et al. (2018), which reweight neighborhoods to emphasize salient relations (e.g., the contrast signaled by $but$) while suppressing noise. In parallel, we encode the text with a pretrained LLM. The resulting hybrid representation—fusing graph-level and text-level features—combines explicit relational structure with rich contextualization.

To empirically evaluate our framework, we conducted extensive experiments on five benchmark datasets spanning biomedical and commonsense reasoning tasks. Results demonstrate that IntuitiveGraphLLM substantially enhances the semantic, structural, and logical understanding of text, producing consistent performance gains over both graph- and text-only baselines. As illustrated in Figure 1, IntuitiveGraphLLM with RoBERTa achieves an average accuracy improvement of **3.85%** (↑) compared to RoBERTa, while IntuitiveGraphLLM variants with DeepSeek and Llama improve by **3.08%** and **3.84%**, respectively, over their base models. IntuitiveGraphLLM not only surpasses the

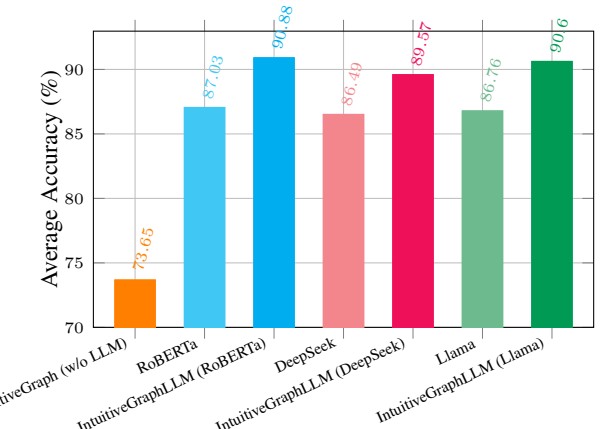

Figure 1: Average accuracy gains of IntuitiveGraphLLM variants (RoBERTa, DeepSeek, Llama) compared to their respective base LLMs and to IntuitiveGraph (w/o LLM), across datasets.

performance of backbone LLMs but also outperforms the structure-aware graph model without LLM integration Bugueño & de Melo (2023) by **7.57%** (↑) on commonsense reasoning benchmarks, indicating the importance of fusing IG-based features with pretrained contextual representations. Moreover, for biomedical reasoning on PubMedQA, IntuitiveGraphLLM outperforms a recent state-of-the-art method Tian et al. (2024) by a significant margin, highlighting IntuitiveGraphLLM's ability to generalize to domain-specific reasoning tasks. To summarize, our main contributions are:

- **Intuitive Graphs (IGs).** We formalize graphs (representation of text) that combine *structural context* (order/proximity/position) with *conceptual relevance* (semantic affinity), producing sparse, interpretable, and human-plausible structures.

- **IntuitiveGraphLLM.** We propose a framework that uses IGs with GATs for the graph branch and an LLM for the text branch; their embeddings (or features) are fused to improve robustness and precision on downstream tasks.

- **Comprehensive study.** We report results across five diverse datasets, multiple IG constructions and node-embedding choices, varied GAT settings, and LLMs, with ablations isolating the effects of semantic gating, graph processing, and fusion.

## 2 BACKGROUND

Existing approaches to graphified text—such as BoW/TF-IDF graphs and sequential or positional schemes—typically connect tokens based on *structure alone* (co-occurrence, order, position) Qian et al. (2024); Toroghi et al. (2024); Tian et al. (2024). These graphs often over-connect function

words and adjacent tokens that are not conceptually related, injecting noise that weakens downstream reasoning with LLMs. We posit that graphs should retain edges only when they are both *contextually* and *conceptually* relevant, and we operationalize this via a *semantic gate* on top of standard structure-aware graph construction.

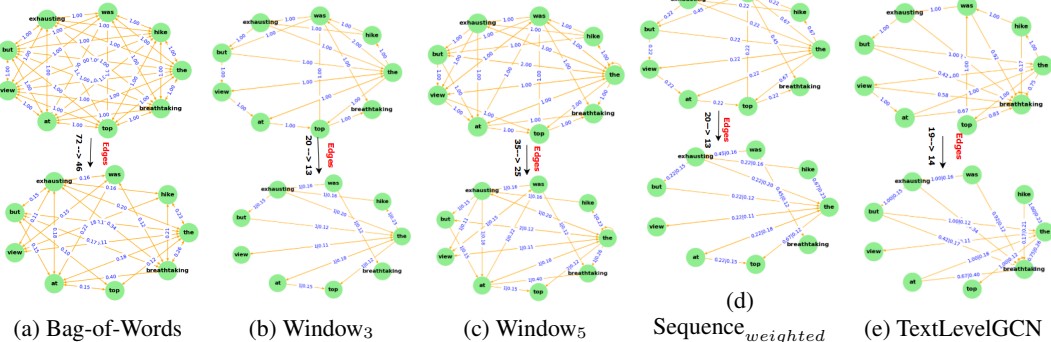

(a) Bag-of-Words    (b) Window$_3$    (c) Window$_5$    (d) Sequence$_{weighted}$    (e) TextLevelGCN

Figure 2: Graph representations of the sentence *"The hike was exhausting, but the view at the top was breathtaking."* From left to right: (a) Bag-of-Words (BoW), (b) Window$_3$, (c) Window$_5$, (d) Sequence$_{weighted}$, and (e) TextLevelGCN. The top row uses structure-aware edges (co-occurrence/order/position). The bottom row shows the corresponding IGs.

Figure 2 contrasts conventional graphs with their IG counterparts for the sentence *"The hike was exhausting, but the view at the top was breathtaking."* All graphs are directed over the *nine unique tokens* in the sentence. IGs are derived by pruning structure-aware edges through a semantic gate, which retains only those edges whose token-embedding cosine similarity exceeds the threshold (e.g., threshold $\tau = 0.3$). Across five schemes, IGs reliably remove spurious links while keeping conceptually plausible edges: *BoW* drops from 72 to 46 edges ($-36.1\%$); Window$_3$ from 20 to 13 ($-35.0\%$); Window$_5$ from 35 to 25 ($-28.6\%$); Sequence$_{weighted}$ from 20 to 13 ($-35.0\%$); and TextLevelGCN from 19 to 14 ($-26.3\%$). On average, IGs reduce edge count by $\approx 32\%$ across these families. Many removed edges involve weakly informative linkages among function or high-frequency words (e.g., the $\rightarrow$ was, but $\leftrightarrow$ view), which are structurally adjacent yet semantically unaligned.

## 3 METHODOLOGY

We decompose IntuitiveGraphLLM framework into four components, each formalized in the following subsections: ($i$) IG construction and initialization (§3.1), ($ii$) IG processing with GATs and graph features (§3.2), ($iii$) Text embedding with a pretrained LLM (§3.3), and ($iv$) Feature fusion (§3.4). Figure 3 summarizes the overall pipeline and information flow.

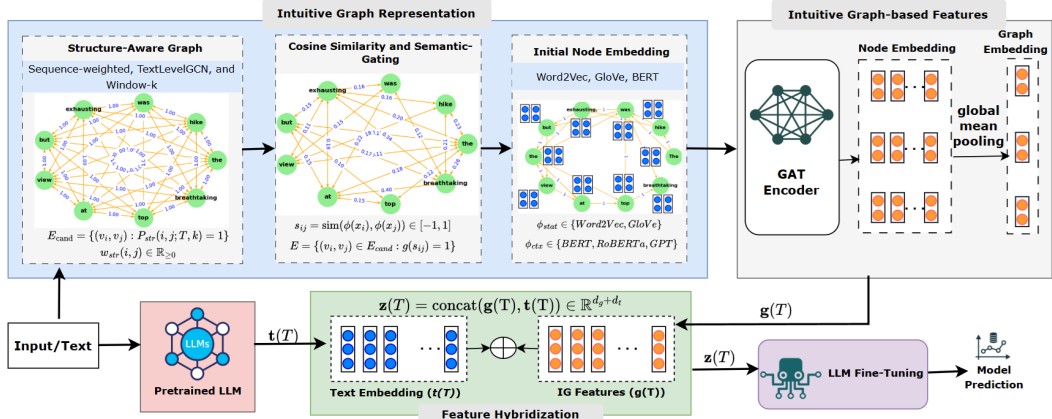

Figure 3: Overview of **IntuitiveGraphLLM**. Text is converted to an IG by forming structural candidate edges and pruning them via semantic-gating; nodes are initialized and encoded with a GAT to produce graph features ($g(T)$), which are fused with LLM-based contextual embeddings ($t(T)$) for end-to-end training and prediction.

## 3.1 Constructing Intuitive Graph and Initializing Node Embeddings

The IG representation of text is designed to effectively capture the structural, contextual, and semantic relationships within an input sequence $T = \{x_1, x_2, \ldots, x_n\}$, where each $x_i$ denotes a word (or token) in the sequence. Algorithm 1 is a pseudocode for IG construction and node feature initialization. An IG augments a structure-aware graph development concept (discussed in Appendix A) with a semantic gate. Let $E_{\text{cand}}$ be edges of a structural scheme (e.g., $\text{Window}_k$, BoW, $\text{Sequence}_{weighted}$, and TextLevelGCN). Edges are first built by a structural predicate $P_{\text{str}}(i, j; T, \kappa) \in \{0, 1\}$ controlled by a scheme $\kappa$:

$$E_{\text{cand}} = \{(v_i, v_j) \,:\, P_{\text{str}}(i, j; T, \kappa) = 1\},$$
$$w_{\text{str}}(i, j) \in \mathbb{R}_{\geq 0}. \tag{1}$$

where $w_{\text{str}}$ is a structural weight (count/TF–IDF/position). Let $\phi(\cdot) \in \mathbb{R}^{d_e}$ be an embedding function and $\text{sim}(\mathbf{u}, \mathbf{v}) = \frac{\mathbf{u}^\top \mathbf{v}}{\|\mathbf{u}\| \, \|\mathbf{v}\|}$ be the cosine similarity. For each $(v_i, v_j) \in E_{\text{cand}}$ compute

$$s_{ij} \;=\; \text{sim}\big(\phi(x_i),\, \phi(x_j)\big) \in [-1, 1]. \tag{2}$$

Edges are retained by a policy $g$ (e.g., threshold values)

$$E \;=\; \{(v_i, v_j) \in E_{\text{cand}} : g(s_{ij}) = 1\}. \tag{3}$$

Applying the semantic gating policy $g$ removes structurally adjacent but semantically weak links (e.g., function-word ties), yielding sparser, more faithful neighborhoods. Next, we leveraged both static and contextual embedding methods to initialize the node embeddings. Let $\phi_{\text{stat}}$ be a static embedding (e.g., GloVe, Word2Vec) and $\phi_{\text{ctx}}(T)_i$ a contextual embedding for token $x_i$ from a pretrained model (e.g., BERT). We form the initial node features $\mathbf{T}^{(0)} = [\mathbf{x}_1^{(0)}; \ldots; \mathbf{x}_n^{(0)}] \in \mathbb{R}^{n \times d}$ through static $(\mathbf{x}_i^{(0)} = \mathbf{P}\,\phi_{\text{stat}}(x_i))$ and contextual $(\mathbf{x}_i^{(0)} = \mathbf{P}\,\phi_{\text{ctx}}(T)_i)$ embedding, where $\mathbf{P} \in \mathbb{R}^{d \times (\cdot)}$ is a learned projection.

---

**Algorithm 1** Constructing IGs and Initializing Node Embeddings

---

**Require:** Input sequence $T = (x_1, x_2 \ldots, x_n)$; structural-aware graph scheme $\kappa$; embedding $\phi$; static $\phi_{\text{stat}}$ and/or contextual $\phi_{\text{ctx}}$; semantic threshold $\tau$; projection $\mathbf{P}$.
**Output:** $\mathcal{IG} = (V, E, W), \mathbf{T}^{(0)}$
 1: $V \leftarrow \{v_1, \ldots, v_n\}$.
 2: Build structural candidates and weights: $E_{\text{cand}} \leftarrow \{(v_i, v_j) : P_{\text{str}}(i, j; T, \kappa) = 1\}, \quad w_{\text{str}}(i, j) \in \mathbb{R}_{\geq 0}$.
 3: Precompute embeddings once:
    $\mathbf{e}_i \leftarrow \phi(x_i) \in \mathbb{R}^{d_e}$ for $i = 1, \ldots, n$.
 4: For each $(v_i, v_j) \in E_{\text{cand}}$, compute cosine $s_{ij} \leftarrow \frac{\mathbf{e}_i^\top \mathbf{e}_j}{\|\mathbf{e}_i\| \|\mathbf{e}_j\|}$.
 5: Apply semantic gate (threshold):
    $E \leftarrow \{(v_i, v_j) \in E_{\text{cand}} : s_{ij} \geq \tau\}$.
 6: **if** $E = \varnothing$ **then**                                            ▷ optional safeguard
 7:     Keep $(v_p, v_q) = \arg\max_{(v_i, v_j) \in E_{\text{cand}}} s_{ij}$
 8:     $E \leftarrow \{(v_p, v_q)\}$.
 9: **end if**
10: Set edge weights:
    $W(i, j) \leftarrow w_{\text{str}}(i, j)$ for all $(v_i, v_j) \in E$.
11: Initialize node embeddings:

$$\mathbf{x}_i^{(0)} \leftarrow \begin{cases} \mathbf{P}\,\phi_{\text{stat}}(x_i), \\ \mathbf{P}\,\phi_{\text{ctx}}(T)_i, \end{cases} \quad i = 1, \ldots, n.$$

12: **return** $(V, E, W), \mathbf{T}^{(0)} = [\mathbf{x}_1^{(0)}; \ldots; \mathbf{x}_n^{(0)}]$.

---

## 3.2 Intuitive Graph Processing with GATs and Graph Features

Let the semantically gated IG be $G = (V, E, W)$ with $|V| = n$. From Section 3.1 we obtain initial node embeddings $\mathbf{T}^{(0)} \in \mathbb{R}^{n \times d_{\text{in}}}$. The semantic gate fixes the (directed) neighborhood

$\mathcal{N}(i) = \{\, j : (v_i, v_j) \in E \,\}$, and all attention is computed *only* over $\mathcal{N}(i)$ (i.e., pruned edges never receive attention weight). To maintain a clean analysis aligned with the IG objective, this variant does not incorporate edge weights $W$ into the attention mechanism. Instead, structural information is conveyed solely through the masked adjacency matrix $E$.

**Layered GAT architecture and dimensions.** We stack $L$ graph-attention layers with hidden width $d_h$. Hidden layers use $K$ heads with concatenation; the final layer uses one head without concatenation. Let $\mathbf{H}^{(\ell)} = [\mathbf{h}_1^{(\ell)}; \ldots; \mathbf{h}_n^{(\ell)}]$ denote node states at depth $\ell$, with $\mathbf{H}^{(0)} = \mathbf{T}^{(0)}$. The dimensionality evolves as $\mathbf{H}^{(1)} \in \mathbb{R}^{n \times (K d_h)}$, $\mathbf{H}^{(\ell+1)} \in \mathbb{R}^{n \times (K d_h)}$ $(1 \le \ell \le L-2)$, $\mathbf{H}^{(L)} \in \mathbb{R}^{n \times d_{\text{out}}}$. A terminal linear projection $\mathbf{P} \in \mathbb{R}^{d_{\text{out}} \times d_{\text{out}}}$ stabilizes the output:

$$\mathbf{Z} = \mathbf{H}^{(L)} \mathbf{P}^\top \in \mathbb{R}^{n \times d_{\text{out}}}. \tag{4}$$

**Masked multi-head attention.** Fix a layer $\ell$ and head $m$. Let $\mathbf{W}^{(\ell,m)}$ and $\mathbf{a}^{(\ell,m)}$ be learnable parameters. For $j \in \mathcal{N}(i)$, define attention logits and the masked softmax as

$$e_{ij}^{(\ell,m)} = \text{LeakyReLU}\Big( \mathbf{a}^{(\ell,m)\top} \big[ \mathbf{W}^{(\ell,m)} \mathbf{h}_i^{(\ell)} \,\big\|\, \mathbf{W}^{(\ell,m)} \mathbf{h}_j^{(\ell)} \big] \Big), \tag{5}$$

$$\alpha_{ij}^{(\ell,m)} = \frac{\exp\big(e_{ij}^{(\ell,m)}\big)}{\sum_{l \in \mathcal{N}(i)} \exp\big(e_{il}^{(\ell,m)}\big)}, \quad j \in \mathcal{N}(i). \tag{6}$$

Edges removed by the semantic gate do not appear in $\mathcal{N}(i)$ and therefore receive no attention mass.

**Message passing, multi-head aggregation, and nonlinearity.** With $\sigma = \text{ReLU}$, hidden layers aggregate per head and concatenate. The final layer aggregates with a single head (no concatenation):

$$\tilde{\mathbf{h}}_i^{(\ell+1)} = \big\|_{m=1}^{K} \sum_{j \in \mathcal{N}(i)} \alpha_{ij}^{(\ell,m)} \mathbf{W}^{(\ell,m)} \mathbf{h}_j^{(\ell)}, \; \mathbf{h}_i^{(\ell+1)} = \sigma\big(\tilde{\mathbf{h}}_i^{(\ell+1)}\big), \; (0 \le \ell \le L-2), \tag{7}$$

$$\mathbf{h}_i^{(L)} = \sigma\left( \sum_{j \in \mathcal{N}(i)} \alpha_{ij}^{(L-1,1)} \mathbf{W}^{(L-1,1)} \mathbf{h}_j^{(L-1)} \right), \qquad \mathbf{H}^{(L)} = [\mathbf{h}_1^{(L)}; \ldots; \mathbf{h}_n^{(L)}]. \tag{8}$$

Equations 5–8 implement a GAT stack with multi-head hidden layers, single-head output, and ReLU after every layer—precisely the behavior of the module MULTILAYERGAT (hidden: $K$ heads with concatenation; output: one head, no concatenation; final linear projection).

**Graph embedding.** Global mean pooling is applied to obtain the final node embeddings (or features), which are subsequently fused with the pretrained LLM representations.

$$\mathbf{g}(T) = \text{MeanPool}(\mathbf{Z}) = \frac{1}{n} \sum_{i=1}^{n} \mathbf{z}_i \in \mathbb{R}^{d_{\text{out}}}, \tag{9}$$

The IG mask reduces neighborhood size and degree variance, tightening the normalization in equation 6 and mitigating attention dilution from function-word ties. Multi-head hidden layers increase representational diversity on sparse IGs; the single-head output fixes the final width $d_{\text{out}}$ for stable fusion. All structure flows through the masked adjacency $E$; weights $W$ are not used inside attention in this variant, keeping the analysis clean and aligned with the IG objective.

### 3.3 Text Embedding with Pretrained LLM

In parallel to the IG branch, we obtain text representations from a pretrained LLM backbone. Given a tokenized input $T = (x_1, \ldots, x_n)$ with subword length $M$, the last hidden layer is $\mathbf{H}^{(L)} = \text{LLM}(T) \in \mathbb{R}^{M \times d_t}$. We extract a sequence embedding through a backbone-fit pooling function $p(\cdot)$:

$$\mathbf{t}(T) = p\Big( \mathbf{H}^{(L)} \Big), \quad p(\mathbf{H}) = \begin{cases} \mathbf{H}_0^{(L)} & \text{-encoder-only (BERT/RoBERTa)} \\ \frac{1}{|S|} \sum_{m \in S} \mathbf{H}_m^{(L)} & \text{-decoder-only (LLaMA/DeepSeek)} \end{cases}$$

$\mathbf{t}(T)$ is a compact, contextual summary of the entire sequence (e.g., the contrast signaled by "*but*" is encoded bidirectionally by the transformer). The text representation is then concatenated with the graph-level features in the fusion head (Section 3.4) and optimized end-to-end with the task loss.

### 3.4 HYBRID FEATURE REPRESENTATION WITH IG- AND LLM-BASED FEATURES

In our proposed framework, a hybrid feature representation is constructed for the model. Let $\mathbf{g}(T) \in \mathbb{R}^{d_g}$ represent the *graph-level* features obtained from the IG branch (Section 3.2) and $\mathbf{t}(T) \in \mathbb{R}^{d_t}$ denote the *text-level* features generated by the LLM (Section 3.3). We build a hybrid feature representation through concatenation:

$$\mathbf{z}(T) = \mathrm{concat}\big(\mathbf{g}(T), \mathbf{t}(T)\big) \in \mathbb{R}^{d_g + d_t}. \tag{10}$$

In our minimal implementation, we use the identity and fuse directly as in equation 10. A linear classifier maps the hybrid vector to logits for $C$ classes:

$$\boldsymbol{\ell}(T) = \mathbf{W}\,\mathbf{z}(T) + \mathbf{b}, \qquad \mathbf{W} \in \mathbb{R}^{C \times (d_g + d_t)},\ \mathbf{b} \in \mathbb{R}^C. \tag{11}$$

Given a labeled dataset $\mathcal{D}_{\mathrm{down}} = \{(T_j, y_j)\}_{j=1}^M$, we minimize the standard cross-entropy:

$$\mathcal{L}_{\mathrm{down}} = -\frac{1}{M} \sum_{j=1}^{M} \log \frac{\exp(\ell_{y_j}(T_j))}{\sum_{c=1}^{C} \exp(\ell_c(T_j))}. \tag{12}$$

The above implementation corresponds to concatenating the text- and graph-level features, followed by an FC layer. A multi-layer MLP head (with nonlinearity, dropout, or normalization) is a drop-in generalization of equation 11. The fusion in equation 10 preserves *complementarity*: emphasizes *relational* cues distilled by the IG mask and GAT, whereas $\mathbf{t}(T)$ captures *global* context and lexical nuance from the LLM. Concatenation preserves both signals without requiring token-level alignment, keeps the head lightweight, and empirically stabilizes training when the two branches (graph and text) differ in scale or domain.

## 4 EXPERIMENTAL SETTINGS

### 4.1 DATASETS

Prior graphified text methods are often tailored to one task or to narrow text regimes Yao et al. (2019), making robustness claims difficult to assess. Our selection stresses ($i$) *domain shift* (consumer, entertainment, general news, political news, biomedical), ($ii$) *document length* (short app reviews vs. long IMDB/HND articles), and ($iii$) *class balance*, to probe whether semantic gating in IGs consistently reduces structural noise and benefits downstream learning. To test whether IntuitiveGraphLLM transfers beyond a single domain or document style, we deliberately evaluate on five public corpora that vary in genre, length, and label skew, spanning sentiment, topic, ideology, and biomedical reasoning. Concretely, we use: App Reviews Grano et al. (2017) (user reviews; sentiment), IMDB Maas et al. (2011) (long-form movie reviews; sentiment), BBC News Greene & Cunningham (2006) (news articles; topic classification), Hyperpartisan News Detection (HND) Kiesel et al. (2018) (news articles), and PubMedQA Jin et al. (2019) (biomedical question answering; we use the artificial subset and its yes/no labels as in our experiments). Table 1 summarizes their statistics, including average token length (ATL), class counts, and length distribution. Additional dataset details are provided in Appendix B.

Table 1: Summary statistics of the datasets. The table reports the number of classes, ATL, and proportions of documents exceeding 100, 512, and 1024 tokens.

| Dataset | Class | ATL | $\geq$ **100** | $\geq$ **512** | $\geq$ **1024** |
|---|---|---|---|---|---|
| PubMedQA | 2 | 27.60 | 0 % | 0 % | 0 % |
| BBC News | 5 | 459.34 | 100 % | 31.22 % | 2.27 % |
| App Review | 5 | 17.44 | 1.51 % | 0 % | 0 % |
| IMDB | 2 | 313.98 | 93.02 % | 14.92 % | 2.22 % |
| HND | 2 | 1203.73 | 98.10 % | 72.93 % | 41.55 % |

### 4.2 MODEL SETTINGS

We configure IntuitiveGraphLLM by systematically varying its core components. ($i$) **Intuitive Graph methods.** We employ four IG constructions—Window$_3$, Window$_5$, Sequence$_{weighted}$, and

TextLevelGCN. For TextLevelGCN, we vary the $n$-gram size from 1 to 3 to capture dependencies at different contextual ranges. During IG construction, we apply semantic gating with a cosine-similarity threshold $\tau \in [0.2, 0.5]$, ensuring that edges are retained only when both structurally relevant and semantically meaningful. Dataset-specific threshold estimates are reported in Appendix D. ($ii$) **Node embeddings.** Initial node representations are derived from three embedding schemes: BERT (768 dimensions), Word2Vec (300 dimensions), and GloVe (300 dimensions). ($iii$) **Graph encoder.** Graph embeddings are produced using a GAT with 2–4 hidden layers, each configured with 4 attention heads, and 128-dimensional hidden/output size. A *global mean pooling* layer aggregates node representations into graph-level embeddings. ($iv$) **LLM backbone.** For the text branch, we experiment with RoBERTa (roberta-base), DeepSeek (DeepSeek-R1-Distill-Qwen-1.5b), and LLaMA (Llama-3.2-1B) to provide contextual sequence-level representations. ($v$) **Optimization.** Models are trained with the Adam optimizer (learning rate $1 \times 10^{-5}$) and cross-entropy loss. These components—IG construction, node embeddings (NE), GAT encoders, and pretrained LLM backbones—are combined to form a unified hybrid framework for robust text representation learning. Further implementation and evaluation details are provided in Appendix C.

## 5 RESULTS

We evaluate the effectiveness of IntuitiveGraphLLM through extensive experiments on five benchmark datasets, systematically varying IG constructions, node embeddings, GAT depths, and LLM backbones. As a starting point, we focus on the RoBERTa-based variant, which consistently demonstrates strong performance across domains. IntuitiveGraphLLM (RoBERTa) achieves high Acc scores: 94.73 on PubMedQA, 72.39±0.5 on App Review, 99.19 on BBC News, 92.31 on HND, and 95.76 on IMDB (Tables 16 and 17 in the Appendix). While these accuracy levels are competitive, performance on PubMedQA and App Review reveals comparatively lower macro $\mathbf{F1}_{ma}$ values, reflecting the effect of class imbalance. As shown in Figure 4, this trend persists across all datasets: macro $\mathbf{F1}_{ma}$ scores are consistently lower than weighted $\mathbf{F1}_{wg}$, underscoring the skewed class distributions.

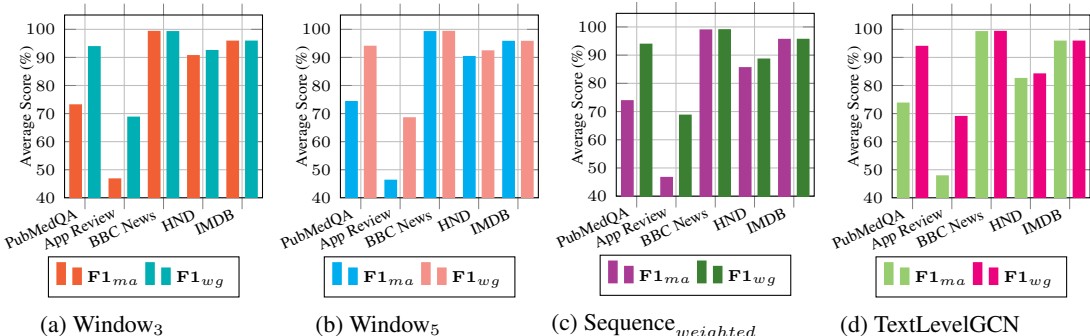

| (a) Window$_3$ | (b) Window$_5$ | (c) Sequence$_{weighted}$ | (d) TextLevelGCN |

Figure 4: Comparison of the average $\mathbf{F1}_{ma}$ and $\mathbf{F1}_{wg}$ scores using different IG methods, GAT layers, and the RoBERTa LLM across all datasets.

### 5.1 CAN INTUITIVEGRAPHLLM IMPROVE THE PERFORMANCE OF BASELINE LLMS?

We compared our best-performing IntuitiveGraphLLM variants against strong baselines, including BERT, RoBERTa, DeepSeek, and Llama (Table 2). Across all datasets, IntuitiveGraphLLM consistently matches or surpasses the performance of its corresponding backbone models. On PubMedQA, IntuitiveGraphLLM improves accuracy by +0.12%, +0.18%, and +0.77% over RoBERTa, DeepSeek, and Llama, respectively. Macro $\mathbf{F1}_{ma}$ scores also increase, with relative gains of +1.16% (RoBERTa), +1.55% (DeepSeek), and +4.97% (Llama). For the more challenging App Review dataset, where severe class imbalance suppresses $\mathbf{F1}_{ma}$, IntuitiveGraphLLM still provides consistent improvements. Notably, the IntuitiveGraphLLM variant (Llama) achieves gains of +3.39% in accuracy and +2.43% in $\mathbf{F1}_{ma}$ over Llama. On BBC News, which is relatively balanced, IntuitiveGraphLLM delivers substantial improvements. With RoBERTa, it yields +2.45% in accuracy and +2.48% in $\mathbf{F1}_{ma}$, while the Llama-based variant achieves a +1.63% accuracy gain. The most pronounced gains occur on the HND dataset, which contains long, complex political news articles. Here, IntuitiveGraphLLM improves accuracy by +10.78% to +12.31% and boosts $\mathbf{F1}_{ma}$ by +10.75% to +15.64%, depending on the backbone. On IMDB, IntuitiveGraphLLM again surpasses all baselines, improving accuracy by

+5.48% over RoBERTa, +0.80% over DeepSeek, and +1.12% over Llama, with similar improvements in $\mathbf{F1}_{ma}$. These results demonstrate that IntuitiveGraphLLM not only enhances accuracy but also yields consistent improvements in macro $\mathbf{F1}_{ma}$ across diverse datasets. The largest benefits are observed on long-document and imbalanced datasets (HND, IMDB, BBC News), underscoring the role of IGs in mitigating noise and highlighting salient relations. The consistent gains across RoBERTa, DeepSeek, and Llama confirm that IG-augmented representation of text provide complementary inductive biases beyond what LLMs achieve alone. We also conducted experiments to assess whether the IG features contain meaningful information that contributes to IntuitiveGraphLLM's performance. The results are presented in Appendix E. Furthermore, a comprehensive comparison and additional details are provided in Appendix I.

Table 2: Performance comparison of IntuitiveGraphLLM and state-of-the-art baseline models across the datasets.

| Model | PubMedQA | | App Review | | BBC News | | HND | | IMDB | |
|---|---|---|---|---|---|---|---|---|---|---|
| | Acc | $\mathbf{F1}_{ma}$ | Acc | $\mathbf{F1}_{ma}$ | Acc | $\mathbf{F1}_{ma}$ | Acc | $\mathbf{F1}_{ma}$ | Acc | $\mathbf{F1}_{ma}$ |
| BoW MLP | 90.74 | 65.23 | 67.65 | 41.69 | 95.12 | 94.93 | 78.46 | 72.12 | 87.60 | 87.60 |
| BERT | 94.14 | 72.31 | 71.20 | 44.84 | 96.74 | 96.57 | 73.84 | 68.86 | 88.80 | 88.79 |
| RoBERTa | 94.61 | 74.81 | 71.97 | 47.72 | 96.74 | 96.62 | 81.53 | 79.81 | 90.28 | 90.26 |
| IntuitiveGraphLLM | 94.73 | 73.65 | 72.39 | 46.17 | 99.19 | 99.10 | 92.31 | 90.56 | 95.76 | 95.76 |
| Gain$_\triangle$ | ↑ 0.12 | ↓ 1.16 | ↑ 0.42 | ↓ 1.55 | ↑ 2.45 | ↑ 2.48 | ↑ 10.78 | ↑ 10.75 | ↑ 5.48 | ↑ 5.50 |
| DeepSeek | 94.33 | 72.37 | 69.22 | 46.65 | 98.37 | 98.42 | 76.92 | 70.68 | 93.60 | 93.60 |
| IntuitiveGraphLLM | 94.51 | 73.92 | 70.52 | 46.92 | 99.19 | 99.24 | 89.23 | 86.32 | 94.40 | 94.40 |
| Gain$_\triangle$ | ↑ 0.18 | ↑ 1.55 | ↑ 1.30 | ↑ 0.27 | ↑ 0.82 | ↑ 0.82 | ↑ 12.31 | ↑ 15.64 | ↑ 0.80 | ↑ 0.80 |
| Llama | 93.75 | 69.81 | 69.09 | 44.37 | 97.56 | 97.61 | 80.00 | 76.84 | 93.40 | 93.40 |
| IntuitiveGraphLLM | 94.52 | 74.78 | 72.48 | 46.80 | 99.19 | 99.18 | 92.31 | 90.56 | 94.52 | 94.52 |
| Gain$_\triangle$ | ↑ 0.77 | ↑ 4.97 | ↑ 3.39 | ↑ 2.43 | ↑ 1.63 | ↑ 1.57 | ↑ 12.31 | ↑ 13.72 | ↑ 1.12 | ↑ 1.12 |

## 5.2 IMPACT OF INTUITIVE GRAPH REPRESENTATION ON LLM PERFORMANCE

Table 1 summarizes the document length statistics of the five datasets. The HND dataset exhibits the longest documents, with an ATL of 1203.73. BBC News and IMDB also contain relatively long texts (ATLs of 459.34 and 313.98, respectively), whereas PubMedQA and App Review are considerably shorter. When examining length distributions, we find that 41.55% of HND articles exceed at least 1024 tokens, compared to only 2.27% of BBC News and 2.22% of IMDB documents, underscoring the particular challenge of modeling HND. However, to assess the effect of IGs on long-text modeling, we compared baseline LLMs with their IntuitiveGraphLLM counterparts. Figure 5 illustrates these results. With RoBERTa (Fig. 5a), IntuitiveGraphLLM achieves accuracy improvements of +2.45% on BBC News, +10.78% on HND, and +5.48% on IMDB. Using DeepSeek (Fig. 5b), the gain on HND is even larger at +12.31%. Similarly, with Llama (Fig. 5c), IntuitiveGraphLLM improves accuracy by +1.63% on BBC News, +12.31% on HND, and +1.12% on IMDB.

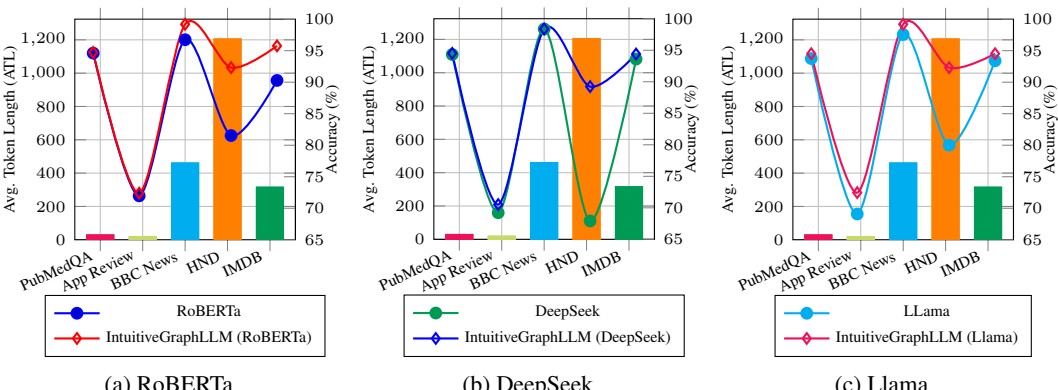

|  (a) RoBERTa | (b) DeepSeek | (c) Llama |

Figure 5: Comparison of performance gains between IntuitiveGraphLLM variants and their baseline LLM counterparts across datasets.

IntuitiveGraphLLM yields a significant average accuracy improvement on HND (+11.80%), IMDB (+2.47%), and BBC News (+1.36%). The obtained results highlight that performance gains are most

pronounced on datasets with longer documents (e.g., HND), where structural noise and semantic drift pose greater challenges for baseline LLMs. Furthermore, consistent improvements on shorter-text datasets (e.g., PubMedQA, App Review) indicate that IG representations also benefit cases with limited context by strengthening structural and semantic alignment. Therefore, integrating IG into LLMs substantially enhances their ability to capture structural and semantic relations, with especially strong effects in long-document settings.

### 5.3 Ablation Study

The IntuitiveGraphLLM framework integrates two complementary components: ($i$) graph-based modules—IG construction, NE, and GATs—and ($ii$) a pretrained LLM for contextual representation. For this ablation study, we focused on RoBERTa and its IG-augmented variant. To quantify the contribution of each part, we performed ablation experiments by systematically removing components, with results reported in Table 3. When the LLM branch is removed (w/o LLM), performance drops sharply across datasets, particularly on App Review (68.23% Acc), BBC News (56.10% Acc), and HND (72.31% Acc), underscoring the im-

Table 3: Results of ablation study

| Dataset | Variant | Acc | $\text{F1}_{ma}$ | $\text{F1}_{wg}$ |
|---|---|---|---|---|
| PubMedQA | w/o LLM | 94.34 | 66.88 | 92.73 |
| | w/o IG | 94.61 | 74.81 | 93.95 |
| | IntuitiveGraphLLM | 94.73 | 73.65 | 93.85 |
| App Review | w/o LLM | 68.23 | 33.72 | 61.10 |
| | w/o IG | 71.97 | 47.72 | 69.04 |
| | IntuitiveGraphLLM | 72.39 | 46.17 | 68.41 |
| BBC News | w/o LLM | 56.10 | 48.10 | 50.94 |
| | w/o IG | 96.74 | 96.62 | 96.76 |
| | IntuitiveGraphLLM | 99.19 | 99.10 | 99.19 |
| HND | w/o LLM | 72.31 | 41.96 | 60.69 |
| | w/o IG | 81.53 | 79.81 | 82.44 |
| | IntuitiveGraphLLM | 92.31 | 90.56 | 92.37 |
| IMDB | w/o LLM | 77.28 | 77.27 | 77.27 |
| | w/o IG | 90.28 | 90.26 | 90.28 |
| | IntuitiveGraphLLM | 95.72 | 95.71 | 95.72 |

portance of contextualized features towards the optimal model. Conversely, when the IG branch is removed (*w/o IG, NE, and GATs*), the LLM alone performs strongly on shorter and moderately long texts—for example, 96.74% on BBC News and 90.28% on IMDB. However, the full IntuitiveGraphLLM consistently attained the top results across all benchmarks. Accuracy improvements over the standalone LLM are +0.12% on PubMedQA, +0.42% on App Review, +2.45% on BBC News, +10.78% on HND, and +5.44% on IMDB. In particular, the gains are largest on long-document datasets such as HND and IMDB, where IG provide structural cues that complement LLM representations.

The ablation study findings highlight several key insights: ($i$) combining IG representations with LLM transforms the framework into a consistently top-performing model; ($ii$) semantic gating and graph-augmented processing benefit for capturing long-range dependencies and mitigating noise, particularly in complex and imbalanced datasets.

## 6 Conclusion

In this paper, we introduced IntuitiveGraphLLM, a hybrid framework that fuses semantically gated Intuitive Graphs with pretrained LLMs to strengthen structural, semantic, and logical understanding of text. By pruning noisy edges and preserving conceptually salient relations, IntuitiveGraphLLM provides interpretable graph structures that complement contextualized embeddings. Extensive experiments on five diverse benchmarks demonstrate consistent accuracy improvements across RoBERTa, LLaMA, and DeepSeek backbones, with average gains of 3.56% and up to 11.00% on long-document datasets such as HND. Comprehensive ablations further confirm that semantic gating and graph–LLM fusion are the key drivers of these improvements. The obtained results highlight IntuitiveGraphLLM as a principled and generalizable approach for enhancing LLMs in both short- and long-text reasoning tasks.

### Use of the Large Language Models (LLMs)

In this paper, LLMs (generative models) were used exclusively for language polishing. Their use was limited to refining the wording in the introduction, methodology, stylizing equations, a few sentences in results, and conclusion. No LLMs were employed in conducting experiments, analyzing results, or drawing conclusions.

## ETHICS STATEMENT

This work complies with the ICLR Code of Ethics. We propose IntuitiveGraphLLM, a framework that constructs Intuitive Graphs (IGs) encoding ($i$) structural context (local order, proximity, position) and ($ii$) conceptual relevance (semantic affinity in embedding space), and fuses IG-based graph features with pretrained LLM representations. All experiments use publicly available datasets; no human subjects, private data, or personally identifiable information are involved. We evaluated IntuitiveGraphLLM in commonsense and biomedical reasoning tasks to demonstrate its effectiveness. The results suggest that IntuitiveGraphLLM can advance the graphical representation of text and language understanding.

## REPRODUCIBILITY STATEMENT

All experiments were run with fixed random seeds, and dataset preprocessing steps, training configurations, and evaluation metrics are fully described in the paper and Appendix. To support reproducibility, we will release the complete source code (including Jupyter notebooks, Python scripts, trained checkpoints, and evaluation results), along with the curated dataset and a README for detailed guidance, either during the rebuttal phase upon request or with the camera-ready submission.

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

# APPENDIX

## A  PRELIMINARIES

In this section, we present notation, contrast conventional *structure-aware* text graphs with Intuitive Graphs (structure and semantics), graph construction, and outline the pretraining and downstream tasks.

**Notation.** Let a text be a token (or span) sequence $T = (x_1, \ldots, x_n)$. A directed graph is $G = (V, E, W)$ with node set $V = \{v_1, \ldots, v_{|V|}\}$, edges $E \subseteq V \times V$, and nonnegative edge weights $W : E \to \mathbb{R}_{\geq 0}$. Node features are $\mathbf{X} \in \mathbb{R}^{|V| \times d}$ with row $\mathbf{x}_i$ for node $v_i$; $\mathbf{A} \in \{0,1\}^{|V| \times |V|}$ denotes the (unweighted) adjacency. Let $\phi(\cdot) \in \mathbb{R}^{d_e}$ be an embedding function and $\mathrm{sim}(\mathbf{u}, \mathbf{v})$ the cosine similarity.

**Structure-aware text graphs.** Traditional text graphs instantiate edges from *structural predicates* only, e.g., co-occurrence within a sliding window, sequential adjacency, or positional heuristics. Formally, given a predicate $P_{\mathrm{str}}(i, j; T) \in \{0, 1\}$, one defines

$$E_{\mathrm{conv}} = \{(v_i, v_j) : P_{\mathrm{str}}(i, j; T) = 1\}$$

$$W_{\mathrm{conv}}(v_i, v_j) \in \mathbb{R}_{\geq 0},$$

where $W_{\mathrm{conv}}$ is typically a count, TF–IDF, distance decay, or a constant. This family captures *where* tokens co-occur (order/proximity/position) but ignores *what* they mean, often over-connecting unrelated nodes and propagating noise.

**Intuitive Graphs.** An Intuitive Graph (IG) augments a structure-aware graph development concept with a semantic gate. Let $E_{\mathrm{cand}}$ be edges of a structural scheme (e.g., $\mathrm{Window}_k$, $\mathrm{Sequence}_{weighted}$, and TextLevelGCN). For each $(v_i, v_j) \in E_{\mathrm{cand}}$, compute

$$s_{ij} = \mathrm{sim}\big(\phi(x_i), \phi(x_j)\big) \in [-1, 1] \tag{13}$$

and retain the edge only if it passes a policy $g$ (threshold, percentile, or top-$K$).

$$\begin{aligned} E &= \{(v_i, v_j) \in E_{\mathrm{cand}} : g(s_{ij}) = 1\}, \\ W(v_i, v_j) &= w_{\mathrm{str}}(i, j) \cdot h(s_{ij}) \end{aligned} \tag{14}$$

where $w_{\mathrm{str}}$ is a structural weight (count/TF–IDF/position) and $h$ optionally reweights by semantics (e.g., $h(s) = \max(s, 0)$ or $h(s) = s$). IGs therefore encode edges that are both *contextual* (licensed by structure) and *conceptual* (supported by semantics), producing sparse, human-plausible graphs.

**Semantic Gating.** The semantic gate is principled and statistically estimated, rather than hand-picked. It actively modifies the graph topology before information propagation and remains enforced throughout the attention process, resulting in measurable sparsity, reduced noise, and consistent downstream gains across different graph constructions and LLM backbones. Moreover, since our text embeddings incorporate all tokens in the sequence, the semantic gate ensures that only the most relevant tokens, both semantically and structurally, are propagated through the Intuitive Graph, preserving stronger signals for text understanding and reasoning.

We used the cosine-similarity method (13) to measure the similarity between embeddings. Cosine similarity-based gating differs Zhang et al. (2025) fundamentally from prior semantic reweighting/pruning methods that (*a*) retain all edges for attention computation, (*b*) rely on heuristic or fixed cutoffs, or (*c*) entangle semantic information directly into attention scores rather than into the adjacency structure itself. Moreover, traditional reweighting mechanisms merely adjust edge strengths while keeping all connections intact, which often results in dense graphs with noisy or redundant relations.

**Graph construction.** Given $T$, we use three modular steps: (*i*) **Node creation** $V = f_{\mathrm{nodes}}(T)$: tokens, merged spans, or task-specific units; (*ii*) **Edge development** $E_{\mathrm{cand}} = f_{\mathrm{edges}}(T; \kappa)$ via a structural scheme $\kappa$ such as $\mathrm{Window}_k$, $\mathrm{Sequence}_{weighted}$, or TextLevelGCN; (*iii*) **Semantic gating and weights**: compute $s_{ij}$ by equation 13, apply $g$, and set $W$ via equation 14. Node features $\mathbf{X}$ are initialized with $\phi$ (e.g., static or contextual embeddings).

**Pretraining and downstream tasks.** Given an IG over $(\mathbf{X}, E, W)$, a graph encoder (e.g., multi-head GAT with global pooling) produces a graph representation $\mathbf{g}(T)$, while a pretrained LLM yields a text representation $\mathbf{t}(T)$. We form a *hybrid* vector

$$\mathbf{z}(T) \;=\; \text{concat}\big(\mathbf{g}(T),\, \mathbf{t}(T)\big), \tag{15}$$

and train a pretrained LLM (predictor) $f_\theta$ with a task-appropriate loss (e.g., cross-entropy) on labeled data $\mathcal{D}_{\text{down}} = \{(T_j, y_j)\}_{j=1}^M$:

$$\mathcal{L}_{\text{down}}(\theta) \;=\; \frac{1}{M} \sum_{j=1}^M \ell\big(f_\theta(\mathbf{z}(T_j)),\, y_j\big). \tag{16}$$

This setup lets the model exploit complementary inductive biases: the IG encoder contributes explicit relational structure, while the LLM provides rich contextualization.

# B  DATASETS

We present a comprehensive overview of the datasets employed in our IntuitiveGraphLLM experiments. Each dataset has been meticulously annotated by domain experts and validated by the respective research communities to ensure high-quality labels and reliability. The data was split into 90% for training and 10% for testing.

**PubMedQA.** PubMedQA Jin et al. (2019) is a biomedical question-answering (QA) dataset derived from PubMed abstracts. The primary task involves answering research-related questions with one of two possible responses: yes or no, using information from corresponding abstracts. The dataset consists of 1,000 expert-annotated instances, 61,200 unlabeled samples, and 211,300 artificially generated QA pairs. Each instance in PubMedQA comprises: ($i$) A question, either the title of an existing research article or one derived from it. ($ii$) A context, which includes the abstract excluding the conclusion. ($iii$) A long answer, corresponding to the abstract's conclusion, which presumably addresses the research question. ($iv$) A yes/no label summarizing the conclusion. For our study, we utilized only the question and the summary answer fields. PubMedQA represents the first QA dataset specifically designed for reasoning over biomedical research texts, particularly those involving quantitative content.

**App Reviews.** This dataset consists of 288,065 English-language user reviews collected from Android applications across 23 categories Grano et al. (2017). The dataset primarily supports fine-grained sentiment analysis in an imbalanced setting, as 60.5% of the total reviews belong to the 4-star rating category. Each review entry includes: ($i$) The application's package name. ($ii$) The user's review text. ($iii$) The date the review was posted. ($iv$) The assigned rating.

**BBC News.** The BBC News dataset Greene & Cunningham (2006) is a publicly available collection of 2,225 English news articles published on the BBC News website between 2004 and 2005. The dataset covers five categories: business, entertainment, politics, sports, and technology. While it exhibits a slight class imbalance—sports being the most represented category with 511 articles and entertainment the least with 386—it remains a valuable resource for topic classification tasks.

**HND.** The HND dataset Kiesel et al. (2018) is designed for binary classification of news articles as hyperpartisan or non-hyperpartisan. It comprises two subsets: by-article and by-publisher, though this study exclusively employs the by-article subset. This subset contains 645 English-language articles labeled via crowdsourcing, with 238 (37%) classified as hyperpartisan and 407 (63%) as non-hyperpartisan. The primary challenge of this dataset lies in detecting hyperpartisan language, which can differ from standard newswriting at multiple linguistic levels, including style, syntax, semantics, and pragmatics.

**IMDB.** The IMDB dataset Maas et al. (2011) contains English-language movie reviews sourced from the Internet Movie Database (IMDB) and is widely used for binary sentiment classification. It consists of 50,000 reviews, with an equal distribution of positive and negative sentiment labels, ensuring a balanced classification setting.

## C EXPERIMENTAL SETTINGS

### C.1 IMPLEMENTATION DETAILS

The experiments are conducted on a system running RHEL 8.8. The hardware configuration included an Intel Ice Lake (Xeon Platinum 8358) (2 sockets *32 cores/socket), 256 GB of RAM, and an NVIDIA A100 80GB PCIe graphics card.

### C.2 EVALUATION METRICS

We evaluate the performance of IntuitiveGraphLLM using standard classification metrics: *accuracy*, *precision*, *recall*, and *F1-score*, reported in both **macro-averaged** and **weighted-averaged** forms Tian et al. (2024); Rahman et al. (2024).

**Performance Gain.** To highlight the improvement of IntuitiveGraphLLM over its base LLM counterparts, we compute the relative gain:

$$\text{Gain}_\Delta = \text{Metric}_{\text{IntuitiveGraphLLM}} - \text{Metric}_{\text{Base LLM}}, \tag{17}$$

where $\text{Metric} \in \{\text{ACC}, \text{F1}_{ma}, \text{F1}_{wg}\}$. Positive (+) values of $\text{Gain}_\Delta$ indicate a performance improvement ($\uparrow$).

## D ESTIMATING THE SEMANTIC GATING THRESHOLD

We estimate the semantic gate threshold $\tau$ using Otsu's method Otsu (1979) and the Benjamini–Hochberg FDR procedure Benjamini & Hochberg (1995). For each document $d$, let $V_d$ denote the token indices with contextual embeddings $\{\mathbf{x}_i \in \mathbb{R}^d\}_{i \in V_d}$, and let $E_{\text{cand}}^{(d)}(K)$ be the set of structure-licensed candidate edges (e.g., *Window-$K$*). Edge cosine similarities are defined as in equation 2. To obtain a dataset-level operating point, we pool edge scores across a subset of documents. Specifically, for each $d \in \mathcal{D}_{\text{sub}}$, we compute $s_{ij}^{(d)}$ for all $(i,j) \in E_{\text{cand}}^{(d)}(K)$ and form $\mathcal{S} = \bigcup_{d \in \mathcal{D}_{\text{sub}}} \{ s_{ij}^{(d)} \}$. We then estimate $\tau$ from $\mathcal{S}$ (via Otsu and FDR).

Table 4: Dataset-level semantic gating threshold ($\tau$) estimation under Window$_3$ and Window$_5$ structure-aware graphs. For each dataset, we report the pooled similarity count $|\mathcal{S}|$ and thresholds using Otsu and FDR approaches.

| Dataset | Window$_3$ | | | Window$_5$ | | |
|---|---|---|---|---|---|---|
| | $\|\mathcal{S}\|$ | $\tau_{Otsu}$ | $\tau_{FDR}$ | $\|\mathcal{S}\|$ | $\tau_{Otsu}$ | $\tau_{FDR}$ |
| HND | 849,714 | 0.4258 | 0.5105 | 1,410,390 | 0.4180 | 0.4991 |
| PubMedQA | 111,096 | 0.4805 | 1.0000 | 177,160 | 0.4648 | 0.6622 |
| BBC News | 1,180,884 | 0.4258 | 0.5343 | 1,960,140 | 0.4023 | 0.5334 |
| App Reviews | 66,468 | 0.5273 | - | 103,730 | 0.5195 | - |
| IMDB | 970,866 | 0.4180 | 0.4675 | 1,610,110 | 0.4102 | 0.4862 |

Table 4 reports the resulting thresholds per dataset: the minimum is $0.4023$ (Otsu, Window$_5$, BBC News) and the maximum is $1.00$ (FDR, Window$_3$, PubMedQA). These estimates guide the choice of a global working threshold.

Table 5: IntuitiveGraphLLM performance using OTSU/FDR-derived $\tau$ values

| Model | Dataset | $\tau$ | Acc | $F1_{wg}$ | $F1_{ma}$ |
|---|---|---|---|---|---|
| **IntuitiveGraphLLM (RoBERTa)** | HND | 0.42 | 92.31 | 92.24 | 90.23 |
| | IMDB | 0.41 | 95.76 | 95.76 | 95.75 |
| | BBC News | 0.42 | 98.37 | 98.39 | 98.22 |
| | App Review | 0.52 | 72.39 | 67.83 | 45.20 |
| | PubMedQA | 0.48 | 94.37 | 93.65 | 73.65 |

We conducted experiments using the dataset-specific $\tau$ values automatically estimated by OTSU and FDR, as shown in Table 5. These thresholds vary across datasets (e.g., $\tau = 0.42$ for BBC News,

$\tau = 0.52$ for App Review). These results are nearly identical to those reported in the manuscript using the default $\tau = 0.3$, and in some cases the difference is within 0.1–0.3%. This demonstrates that the semantic gate is robust to a wide range of $\tau$ values. Our choice of $\tau = 0.3$ was intentionally conservative to avoid over-pruning, but the additional results confirm that performance is not sensitive to moderate deviations in $\tau$.

## E  IMPACT OF THE IG FEATURES

To examine whether the observed gains arise merely from concatenation (*soft prompting*) rather than from meaningful graph information, we performed control experiments where the IG features/embeddings ($\mathbf{g}(T)$) were randomly permuted/shuffled before fusion with the LLM outputs ($\mathbf{t}(T)$).

Table 6 reports the correlation metrics between the original and permuted IG embeddings across datasets. The near-zero values of cosine similarity (CS), mean Pearson (MP), and mean Spearman (MS) correlations confirm that the permutation completely disrupts the semantic structure; the permuted embeddings are orthogonal and uncorrelated with the originals. Figure 6 visualizes IG features for several IMDB samples before and after shuffling.

Table 6: Correlation metrics before vs. after IG feature shuffling

| Dataset | Avg. Cosine Similarity (CS) | Mean Pearson (MP) | Mean Spearman (MS) |
|---------|------------------------------|-------------------|--------------------|
| HND | 0.0013 | 0.0012 | 0.0023 |
| PubMedQA | 0.005 | -0.0091 | 0.0006 |
| BBC News | 0.0203 | 0.0146 | 0.0116 |
| App Review | 0.0128 | 0.0112 | 0.0076 |
| IMDB | 0.0161 | 0.0133 | 0.0106 |

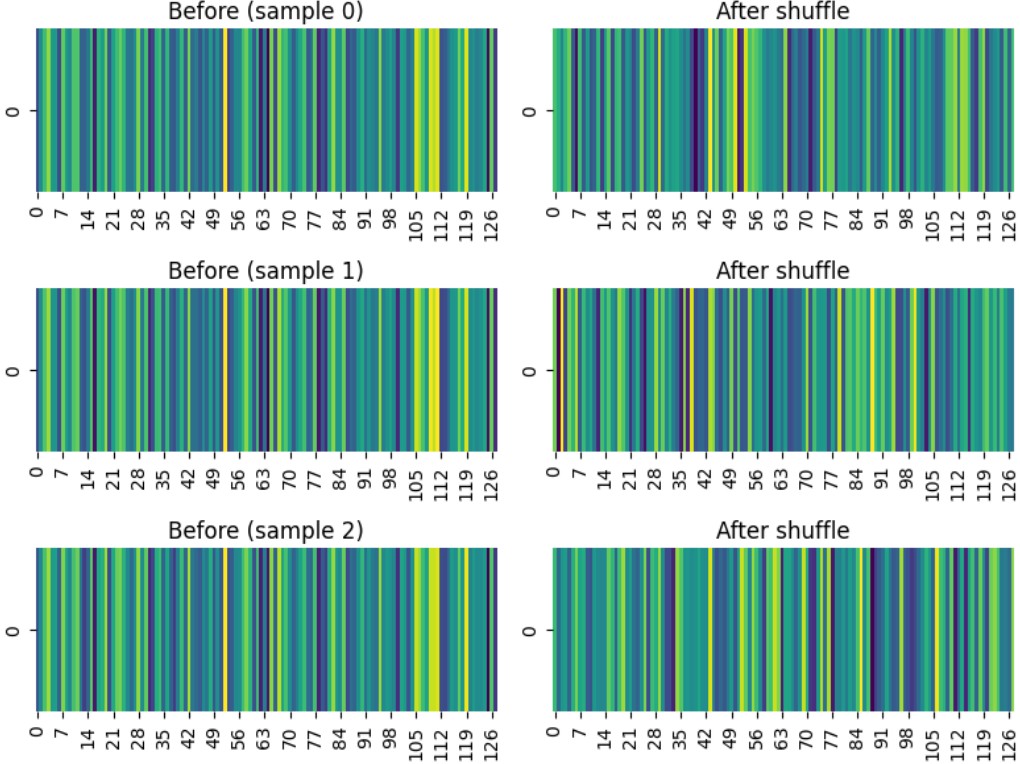

Figure 6: Visualization of IG feature representations for several IMDB samples before and after shuffling.

We then re-trained the IntuitiveGraphLLM (RoBERTa) model using these permuted features. As shown in Table 7, the Acc and $F1_{ma}$ scores dropped by approximately 11% on HND, 0.8% on IMDB, and 2.4% on BBC News. These declines indicate that aligned IG features carry non-trivial semantic signals that aid contextual understanding beyond text embeddings alone. For App Review and PubMedQA, the performance remained similar, likely because their ATL is short (see Table 1), limiting the graph's ability to encode additional structure. Consistent with our ablation results (see Table 3), IG contributes more strongly to long-document datasets, confirming that IntuitiveGraphLLM is especially more effective when semantic and relational dependencies extend across long tokens.

Table 7: Performance comparison before and after shuffling IG features.

| Model | Dataset | No Shuffle | | After Shuffle | | Effect | |
|---|---|---|---|---|---|---|---|
| | | Acc | $F1_{ma}$ | Acc | $F1_{ma}$ | $\Delta$Acc | $\Delta F1_{ma}$ |
| **IntuitiveGraphLLM (RoBERTa)** | HND | 92.31 | 90.56 | 81.54 | 78.90 | -10.77↓ | -11.66↓ |
| | IMDB | 95.76 | 95.76 | 94.96 | 94.96 | -0.80↓ | -0.80↓ |
| | BBC | 99.19 | 99.10 | 96.75 | 96.84 | -2.44↓ | -2.26↓ |
| | App Review | 72.39 | 46.17 | 72.09 | 47.09 | -0.30↓ | +0.92↑ |
| | PubMedQA | 94.73 | 73.65 | 94.52 | 74.12 | -0.21↓ | +0.47↑ |

## F RESULTS OF THE INTUITIVEGRAPHLLM

We conduct comprehensive experiments with IntuitiveGraphLLM (RoBERTa) across the five datasets. Tables 10–14 report accuracy (Acc), macro ($\mathbf{F1}_{ma}$) and weighted ($\mathbf{F1}_{wg}$) under a wide range of IG configurations—Window$_3$, Window$_5$, and Sequence$_{weighted}$—along with varying GAT depths and three node-embedding (NE) schemes (e.g., BERT, Word2Vec, and GloVe). Table 15 presents results for the TextLevelGCN variant across all datasets. Tables 10–14 show the results of IntuitiveGraphLLM for each dataset with different settings. The best-performing configurations and results are summarized in Tables 16 and 17. Overall, the results show consistent performance of IntuitiveGraphLLM (RoBERTa) across datasets. We also evaluate DeepSeek- and LLama-based variants and observe similarly consistent improvements.

## G INTUITIVE GRAPH WITH LLM-BASED EMBEDDING

To examine whether LLM-based embedding models could further improve performance, we conducted additional experiments using Jina-Embeddings v3 (jinaai/jina-embeddings-v3) Günther et al. (2023) under the same IG construction and fusion settings. Table 8 shows the results across datasets. The obtained results are significantly inferior to ($i$) our IntuitiveGraphLLM performance and ($ii$) the baseline LLM backbones (RoBERTa, DeepSeek, Llama). In particular, performance drops sharply on long-document datasets (HND, BBC), where global relational reasoning is most crucial.

Table 8: Performance of IG with Jina Embedding

| Dataset | Acc | $F1_{wg}$ | $F1_{ma}$ |
|---|---|---|---|
| HND | 64.62 | 58.68 | 43.05 |
| IMDB | 89.88 | 89.88 | 89.87 |
| App Review | 68.31 | 63.65 | 41.16 |
| PubMedQA | 93.04 | 90.25 | 53.38 |
| BBC News | 34.96 | 30.97 | 30.85 |

IntuitiveGraphLLM depends on fine-grained token-level contextual representations to construct semantically gated graphs and to perform effective graph–text fusion. However, LLM-based embedding models (e.g., Jina-Embeddings, NV-EmbED) collapse the entire text into a single sentence-level vector, discarding the token-level information required for IG construction, semantic gating, and message passing. As a result, the IG and embedding branches become semantically misaligned, leading to poor fusion and significantly weaker results. In contrast, models like RoBERTa, Llama, and DeepSeek

provide rich token-level contextual embeddings that synergize effectively with IG, especially on long-document datasets. This explains why IntuitiveGraphLLM (RoBERTa/Llama/DeepSeek) outperforms the embedding-based variants.

## H  PERFORMANCE OF THE GRAPH EMBEDDING MODELS

In addition, we conducted experiments with a stronger graph representation model, Graphormer Ying et al. (2021), and GAT under the same settings. Table 9 shows the results. Across all datasets, neither Graphormer nor GAT alone comes close to the performance of any IntuitiveGraphLLM variant. Graphormer performs substantially worse than even the GAT baseline on several datasets (e.g., BBC News). These findings highlight two key points: (i) Graph-only models are struggling with text understanding. Even powerful graph transformers like Graphormer cannot fully model linguistic semantics or long-range textual dependencies in isolation. In contrast, the strength of IntuitiveGraphLLM lies in the fusion of LLM contextual semantics with IG relational structure. The consistent improvements across datasets indicate that IG complements the LLM by introducing structured relational reasoning that neither component achieves alone. These additional results reinforce that the proposed IntuitiveGraphLLM framework is not simply a graph encoder, but an effective hybrid model that leverages both token-level semantics and graph-level relational knowledge.

Table 9: Performance of graph embedding models

| Dataset | Graphormer | | GAT | |
|---|---|---|---|---|
| | Acc | $F1_{ma}$ | Acc | $F1_{ma}$ |
| HND | 72.31 | 41.96 | 72.31 | 41.96 |
| IMDB | 51.92 | 34.18 | 77.28 | 77.27 |
| App Review | 68.04 | 34.79 | 68.23 | 33.72 |
| PubMedQA | 92.86 | 48.15 | 94.34 | 66.88 |
| BBC News | 21.95 | 18.17 | 56.10 | 48.10 |

Table 10: IntuitiveGraphLLM (RoBERTa) performance on PubMedQA dataset

| IG Methods | Node Embed. | #GL | Acc | $F1_{ma}$ | $F1_{wg}$ |
|---|---|---|---|---|---|
| $\text{Window}_3$ | BERT | 2 | 94.40 | 73.79 | 93.71 |
| | | 3 | 94.23 | 73.42 | 93.58 |
| | | 4 | 94.47 | 72.34 | 93.55 |
| | Word2Vec | 2 | 94.67 | 73.02 | 93.75 |
| | | 3 | 94.60 | 74.01 | 93.84 |
| | | 4 | 94.29 | 73.55 | 93.63 |
| | GloVe | 2 | 94.39 | 74.44 | 93.80 |
| | | 3 | 93.83 | 73.21 | 93.36 |
| | | 4 | 93.92 | 73.20 | 93.40 |
| $\text{Window}_5$ | BERT | 2 | 94.58 | 73.88 | 93.82 |
| | | 3 | 93.96 | 73.65 | 93.48 |
| | | 4 | 93.98 | 73.13 | 93.42 |
| | Word2Vec | 2 | 94.58 | 73.39 | 93.74 |
| | | 3 | 94.28 | 74.52 | 93.75 |
| | | 4 | 94.19 | 72.98 | 93.50 |
| | GloVe | 2 | 94.62 | 74.20 | 93.88 |
| | | 3 | 93.75 | 72.56 | 93.23 |
| | | 4 | 93.42 | 72.54 | 93.07 |
| $\text{Sequence}_{weighted}$ | BERT | 2 | 94.63 | 72.60 | 93.66 |
| | | 3 | 93.90 | 73.57 | 93.44 |
| | | 4 | 93.36 | 72.95 | 93.58 |
| | Word2Vec | 2 | 94.49 | 74.57 | 93.86 |
| | | 3 | 93.85 | 72.18 | 93.23 |
| | | 4 | 94.03 | 72.38 | 93.34 |
| | GloVe | 2 | 94.59 | 74.10 | 93.85 |
| | | 3 | 94.16 | 73.13 | 93.51 |
| | | 4 | 94.28 | 72.88 | 93.53 |

Table 11: IntuitiveGraphLLM (RoBERTa) performance on IMDB dataset

| IG Methods | Node Embed. | #GL | Acc | $\mathbf{F1}_{ma}$ | $\mathbf{F1}_{wg}$ |
|---|---|---|---|---|---|
| Window$_3$ | BERT | 2 | 94.80 | 94.80 | 94.80 |
| | | 3 | 95.00 | 95.00 | 95.00 |
| | | 4 | 95.08 | 95.08 | 95.08 |
| | Word2Vec | 2 | 95.36 | 95.36 | 95.36 |
| | | 3 | 94.48 | 94.46 | 94.47 |
| | | 4 | 95.72 | 95.71 | 95.72 |
| | GloVe | 2 | 95.16 | 95.15 | 95.16 |
| | | 3 | 94.68 | 94.68 | 94.68 |
| | | 4 | 94.32 | 94.32 | 94.32 |
| Window$_5$ | BERT | 2 | 95.64 | 95.64 | 95.64 |
| | | 3 | 94.68 | 94.65 | 94.67 |
| | | 4 | 95.16 | 95.15 | 95.16 |
| | Word2Vec | 2 | 94.76 | 94.76 | 94.76 |
| | | 3 | 95.64 | 95.63 | 95.64 |
| | | 4 | 95.64 | 95.63 | 95.64 |
| | GloVe | 2 | 95.32 | 95.32 | 95.32 |
| | | 3 | 94.52 | 94.51 | 94.52 |
| | | 4 | 95.28 | 95.26 | 95.27 |
| Sequence$_{weighted}$ | BERT | 2 | 95.60 | 95.59 | 95.60 |
| | | 3 | 94.92 | 94.92 | 94.92 |
| | | 4 | - | - | - |
| | Word2Vec | 2 | 95.52 | 95.51 | 95.52 |
| | | 3 | 95.24 | 95.24 | 95.24 |
| | | 4 | 94.36 | 94.36 | 94.36 |
| | GloVe | 2 | 95.44 | 95.43 | 95.44 |
| | | 3 | 94.80 | 94.80 | 94.80 |
| | | 4 | 95.00 | 95.00 | 95.00 |

## I    COMPARISON WITH STATE-OF-THE-ART MODELS

We compare each baseline model with its IntuitiveGraphLLM counterpart across five datasets (Table 18 and Figure 7). Overall, IntuitiveGraphLLM matches or surpasses its corresponding backbone in nearly all settings. On **PubMedQA**, the RoBERTa-based variant attains the best accuracy (94.67%). On **App Review**, the LLaMA-based variant yields the highest accuracy (72.48%). On **BBC News** (a relatively balanced dataset), all three IntuitiveGraphLLM variants (RoBERTa, DeepSeek, LLaMA) reach 99.19% accuracy, exceeding their base models. On **HND** (long, complex documents), the RoBERTa- and LLaMA-based variants achieve 92.31% accuracy, while the DeepSeek-based variant underperforms relative to the others. On **IMDB**, the RoBERTa-based model attains the highest accuracy (95.72%). Trends in macro and weighted F1 are consistent with the accuracy improvements. For completeness, we also report training time (minutes) to facilitate comparison.

## J    COMPUTATIONAL AND MEMORY OVERHEAD

While IntuitiveGraphLLM introduces a graph-based branch, the framework was deliberately designed to keep the additional computational and memory overhead lightweight and bounded. Our analyses support this from several perspectives:

**Semantic gating yields highly sparse graphs.** As reported in subsection 4.2, the semantic gate removes 30–35% of candidate edges on average. This produces very sparse adjacency structures, so the GAT complexity scales as: $O(|E|.h)$ rather than as quadratic in the number of tokens. Because the gated IG graphs for long documents remain small graph expansion never becomes a bottleneck.

**The GAT branch is extremely lightweight relative to the LLM.** We computed the parameter distribution across all components (GAT, fusion layer, and RoBERTa). The results show that GAT parameters contribute only 0.19% of total trainable parameters, Fusion/FC layers add ≈0.04%, LLM encoder dominates with >99% of total parameters. Thus, both computational and memory costs of adding IG are very small compared to the LLM forward/backward pass, which remains the dominant factor.

Table 12: IntuitiveGraphLLM (RoBERTa) performance on App Review dataset

| IG Methods | Node Embed. | #GL | Acc | $F1_{ma}$ | $F1_{wg}$ |
|---|---|---|---|---|---|
| Window$_3$ | BERT | 2 | 72.32 | 47.23 | 68.58 |
| | | 3 | 70.17 | 46.42 | 67.64 |
| | | 4 | 69.03 | 45.87 | 66.84 |
| | Word2Vec | 2 | 72.37 | 46.64 | 68.64 |
| | | 3 | 70.47 | 47.32 | 68.03 |
| | | 4 | 70.24 | 46.56 | 67.60 |
| | GloVe | 2 | 71.99 | 46.68 | 68.63 |
| | | 3 | 70.32 | 46.43 | 67.73 |
| | | 4 | 68.97 | 46.67 | 67.27 |
| Window$_5$ | BERT | 2 | 72.16 | 46.56 | 68,49 |
| | | 3 | 69.54 | 45.73 | 67.26 |
| | | 4 | 68.99 | 45.58 | 66.62 |
| | Word2Vec | 2 | 72.31 | 47.80 | 69.15 |
| | | 3 | 70.20 | 46.26 | 67.75 |
| | | 4 | 68.81 | 46.35 | 67.00 |
| | GloVe | 2 | 72.39 | 46.17 | 68.41 |
| | | 3 | 69.80 | 46.32 | 67.63 |
| | | 4 | 69.50 | 46.69 | 67.51 |
| Sequence$_{weighted}$ | BERT | 2 | 72.42 | 46.15 | 68.41 |
| | | 3 | 70.06 | 46.27 | 67.79 |
| | | 4 | 68.81 | 46.97 | 67.36 |
| | Word2Vec | 2 | 72.69 | 47.05 | 68.98 |
| | | 3 | 70.64 | 46.81 | 67.91 |
| | | 4 | 68.69 | 47.42 | 67.37 |
| | GloVe | 2 | 72.49 | 46.48 | 68.54 |
| | | 3 | 70.88 | 46.10 | 67.60 |
| | | 4 | 72.21 | 47.10 | 68.63 |

**The overhead is justified by large gains on long-document datasets.** Although IG introduces modest overhead, it yields substantial performance improvements such as HND: +10.7% to +11.5% (Acc/F1), BBC: +3–4%, and IMDB: +1.5–2.3%. These gains occur precisely when self-attention becomes expensive and less effective. IG offers a sparse, structured mechanism to capture long-range relations, allowing the model to avoid relying solely on full attention over long sequences. Overall, the graph branch contributes less than 1% additional parameters, remains highly efficient due to semantic sparsity, and yields meaningful accuracy improvements on long-document benchmarks (e.g., HND, BBC, and IMDB).

Table 13: IntuitiveGraphLLM (RoBERTa) performance on BBC News dataset

| IG Methods | Node Embed. | #GL | Acc | $F1_{ma}$ | $F1_{wg}$ |
|---|---|---|---|---|---|
| Window₃ | BERT | 2 | 98.37 | 98.34 | 98.38 |
| | | 3 | 99.19 | 99.10 | 99.19 |
| | | 4 | 99.19 | 99.10 | 99.19 |
| | Word2Vec | 2 | 97.56 | 97.43 | 97.57 |
| | | 3 | 99.19 | 99.10 | 99.19 |
| | | 4 | 99.19 | 99.10 | 99.19 |
| | GloVe | 2 | 98.37 | 98.34 | 98.38 |
| | | 3 | 99.19 | 99.10 | 99.19 |
| | | 4 | 98.37 | 98.34 | 98.38 |
| Window₅ | BERT | 2 | 99.19 | 99.10 | 99.19 |
| | | 3 | 99.19 | 99.10 | 99.19 |
| | | 4 | 97.56 | 97.57 | 97.56 |
| | Word2Vec | 2 | 98.37 | 98.32 | 98.38 |
| | | 3 | 98.37 | 98.34 | 98.38 |
| | | 4 | 98.37 | 98.34 | 98.38 |
| | GloVe | 2 | 98.37 | 98.34 | 98.38 |
| | | 3 | 99.19 | 99.10 | 99.19 |
| | | 4 | 98.37 | 98.34 | 98.42 |
| Sequence$_{weighted}$ | BERT | 2 | 96.75 | 96.66 | 96.76 |
| | | 3 | 99.19 | 99.10 | 99.19 |
| | | 4 | 97.56 | 97.30 | 97.58 |
| | Word2Vec | 2 | 97.56 | 97.46 | 97.58 |
| | | 3 | 99.19 | 99.10 | 99.10 |
| | | 4 | 98.37 | 98.22 | 98.39 |
| | GloVe | 2 | 98.37 | 98.22 | 98.39 |
| | | 3 | 98.37 | 98.34 | 98.38 |
| | | 4 | 97.56 | 97.57 | 97.56 |

Table 14: IntuitiveGraphLLM (RoBERTa) performance on HND dataset

| IG Methods | Node Embed. | #GL | Acc | $F1_{ma}$ | $F1_{wg}$ |
|---|---|---|---|---|---|
| Window₃ | BERT | 2 | 86.15 | 83.97 | 86.61 |
| | | 3 | 92.31 | 90.23 | 92.24 |
| | | 4 | 81.53 | 79.39 | 82.36 |
| | Word2Vec | 2 | 87.69 | 82.67 | 86.83 |
| | | 3 | 81.54 | 79.39 | 82.36 |
| | | 4 | 92.31 | 90.56 | 92.37 |
| | GloVe | 2 | 86.15 | 83.97 | 86.61 |
| | | 3 | 90.77 | 88.05 | 90.59 |
| | | 4 | 87.69 | 83.42 | 87.17 |
| Window₅ | BERT | 2 | 89.23 | 86.32 | 89.13 |
| | | 3 | 86.15 | 81.72 | 85.74 |
| | | 4 | 89.23 | 86.32 | 89.13 |
| | Word2Vec | 2 | 89.23 | 85.79 | 88.91 |
| | | 3 | 90.77 | 87.56 | 90.38 |
| | | 4 | 90.77 | 88.84 | 0.91 |
| | GloVe | 2 | 92.31 | 90.23 | 92.24 |
| | | 3 | 89.23 | 85.17 | 89.53 |
| | | 4 | 90.77 | 88.05 | 90.59 |
| Sequence$_{weighted}$ | BERT | 2 | 86.15 | 80.02 | 84.96 |
| | | 3 | 87.69 | 84.07 | 87.46 |
| | | 4 | 87.69 | 84.07 | 87.46 |
| | Word2Vec | 2 | 89.23 | 85.17 | 88.63 |
| | | 3 | 83.08 | 79.85 | 83.45 |
| | | 4 | 80.00 | 76.19 | 80.44 |
| | GloVe | 2 | 89.23 | 87.18 | 89.47 |
| | | 3 | 83.08 | 80.89 | 83.77 |
| | | 4 | 86.15 | 83.52 | 86.46 |

Table 15: The performance of the IntuitiveGraphLLM (RoBERTa) (with TextLevelGCN and GAT (GL #3)) model across all datasets. The $\mathbf{Acc}$, $\mathbf{F1}_{ma}$, and $\mathbf{F1}_{wg}$ are reported.

| Dataset | Node Embed. | 1-gram | | | 2-gram | | | 3-gram | | |
|---|---|---|---|---|---|---|---|---|---|---|
| | | **Acc** | $\mathbf{F1}_{ma}$ | $\mathbf{F1}_{wg}$ | **Acc** | $\mathbf{F1}_{ma}$ | $\mathbf{F1}_{wg}$ | **Acc** | $\mathbf{F1}_{ma}$ | $\mathbf{F1}_{wg}$ |
| PubMedQA | GloVe | 94.66 | 73.73 | 93.83 | 94.05 | 73.26 | 93.47 | 94.31 | 72.38 | 93.48 |
| | Word2Vec | **94.73** | 73.65 | 93.85 | 94.15 | 73.42 | 93.54 | 93.99 | 73.33 | 93.45 |
| | BERT | 94.27 | **74.61** | 93.76 | 94.31 | 73.27 | 93.60 | 93.91 | 73.33 | 93.41 |
| App Review | GloVe | **72.23** | 47.73 | 68.88 | 71.15 | 46.61 | 68.02 | 69.40 | 45.03 | 66.84 |
| | Word2Vec | 72.12 | **48.47** | 69.05 | 70.72 | 47.26 | 68.26 | 68.83 | 46.14 | 67.18 |
| | BERT | 72.22 | 46.99 | 69.06 | 70.43 | 47.12 | 68.09 | 68.81 | 46.81 | 67.28 |
| BBC News | GloVe | 98.37 | 98.34 | 98.38 | 98.37 | 98.34 | 98.38 | 98.37 | 98.34 | 98.38 |
| | Word2Vec | 98.37 | 98.34 | 98.38 | 97.56 | 97.57 | 97.55 | 98.37 | 98.34 | 98.38 |
| | BERT | **99.19** | **99.10** | 99.19 | 99.19 | 99.10 | 99.19 | 98.37 | 99.09 | 98.38 |
| HND | GloVe | 83.08 | 81.99 | 83.15 | 83.08 | 81.31 | 82.81 | 83.08 | 81.31 | 82.81 |
| | Word2Vec | 81.54 | 79.81 | 81.36 | **84.62** | **82.41** | 84.04 | 83.08 | 81.67 | 83.00 |
| | BERT | 83.08 | 81.67 | 83.00 | 73.85 | 73.44 | 74.30 | 81.54 | 80.18 | 81.54 |
| IMDB | GloVe | 94.80 | 94.78 | 94.79 | 95.72 | 95.71 | 95.72 | 95.72 | 95.71 | 95.72 |
| | Word2Vec | 95.24 | 95.23 | 95.24 | **95.76** | **95.76** | 95.76 | 95.36 | 95.36 | 95.36 |
| | BERT | 95.28 | 95.27 | 95.28 | 94.32 | 94.32 | 94.32 | 95.24 | 95.23 | 95.24 |

Table 16: **Top-performing IntuitiveGraphLLM models.** Results using Window$_3$, Window$_5$, and Sequence$_{weighted}$ IG methods with GloVe, Word2Vec, and BERT node embeddings. Semantic thresholds $\tau \in [0.2, 0.5]$, GAT layers (#GL = 2–4), and RoBERTa as the LLM backbone are applied across all datasets.

| Dataset | Node Embed. | #GL | Window$_3$ | | #GL | Window$_5$ | | #GL | Sequence$_{weighted}$ | |
|---|---|---|---|---|---|---|---|---|---|---|
| | | | **Acc** | $\mathbf{F1}_{ma}$ | | **Acc** | $\mathbf{F1}_{ma}$ | | **Acc** | $\mathbf{F1}_{ma}$ |
| PubMedQA | GloVe | 2 | 94.39 | 74.44 | 2 | 94.62 | 74.20 | 2 | 94.59 | 74.10 |
| | Word2Vec | 2 | 94.67 | 73.02 | 2 | 94.58 | 73.39 | 2 | 94.49 | 74.57 |
| | BERT | 4 | 94.47 | 72.34 | 2 | 94.58 | 73.88 | 2 | 94.63 | 72.60 |
| App Review | GloVe | 2 | 71.99 | 46.68 | 2 | 72.39 | 46.17 | 2 | 72.49 | 46.48 |
| | Word2Vec | 2 | 72.37 | 46.64 | 2 | 72.31 | 47.80 | 2 | 72.69 | 47.05 |
| | BERT | 2 | 72.32 | 47.23 | 2 | 72.16 | 46.56 | 2 | 72.42 | 46.15 |
| BBC News | GloVe | 3 | 99.19 | 99.10 | 3 | 99.19 | 99.10 | 3 | 98.37 | 98.34 |
| | Word2Vec | 4 | 99.19 | 99.10 | 3 | 98.37 | 98.34 | 3 | 99.19 | 99.10 |
| | BERT | 3 | 99.19 | 99.10 | 2 | 99.19 | 99.10 | 3 | 99.19 | 99.10 |
| HND | GloVe | 3 | 90.77 | 88.05 | 2 | 92.31 | 90.23 | 2 | 89.23 | 87.18 |
| | Word2Vec | 4 | 92.31 | 90.56 | 4 | 90.77 | 88.84 | 2 | 89.23 | 85.17 |
| | BERT | 3 | 92.31 | 90.23 | 2 | 89.23 | 86.32 | 3 | 87.69 | 84.07 |
| IMDB | GloVe | 2 | 95.16 | 95.15 | 2 | 95.32 | 95.32 | 2 | 95.44 | 95.43 |
| | Word2Vec | 4 | 95.72 | 95.71 | 4 | 95.64 | 95.63 | 2 | 95.52 | 95.51 |
| | BERT | 4 | 95.08 | 95.08 | 2 | 95.64 | 95.64 | 2 | 95.60 | 95.59 |

Table 17: **Top-performing IntuitiveGraphLLM models.** Results using TextLevelGCN with $n$-gram settings (1-$g$, 2-$g$, 3-$g$) and GloVe, Word2Vec, and BERT node embeddings. Semantic thresholds $\tau \in [0.2, 0.5]$, GAT layers (#GL = 3), and RoBERTa as the LLM backbone are applied.

| Dataset | Node Embed. | 1-gram | | 2-gram | | 3-gram | |
|---|---|---|---|---|---|---|---|
| | | **Acc** | $\mathbf{F1}_{ma}$ | **Acc** | $\mathbf{F1}_{ma}$ | **Acc** | $\mathbf{F1}_{ma}$ |
| PubMedQA | Word2Vec | 94.73 | 73.65 | 94.15 | 73.42 | 93.99 | 73.33 |
| App Review | GloVe | 72.23 | 47.73 | 71.15 | 46.61 | 69.40 | 45.03 |
| BBC News | BERT | 99.19 | 99.10 | 99.19 | 99.10 | 98.37 | 99.09 |
| HND | Word2Vec | 81.54 | 79.81 | 84.62 | 82.41 | 83.08 | 81.67 |
| IMDB | Word2Vec | 95.24 | 95.23 | 95.76 | 95.76 | 95.36 | 95.36 |

Table 18: Overall comparison of baselines and their IntuitiveGraphLLM counterparts across datasets: accuracy (Acc), macro/weighted **F1**, and training time (min).

| Dataset | Model | Acc | $\mathbf{F1}_{ma}$ | $\mathbf{F1}_{wg}$ | Time $[min]$ |
|---|---|---|---|---|---|
| PubMedQA | BoW MLP | 90.74 | 65.23 | 90.76 | 41.67 |
| | BERT | 94.14 | 72.31 | 93.38 | 127.25 |
| | RoBERTa | 94.61 | 74.81 | 93.95 | 124.17 |
| | IntuitiveGraphLLM (RoBERTa) | **94.73** | 73.02 | 93.75 | 46.84 |
| | DeepSeek | 94.33 | 72.37 | 93.49 | 314.37 |
| | IntuitiveGraphLLM (DeepSeek) | 94.51 | 73.92 | 93.79 | 386.96 |
| | Llama | 93.75 | 69.81 | 92.86 | 225.44 |
| | IntuitiveGraphLLM (Llama) | 94.52 | 74.78 | 93.91 | 183.77 |
| App Review | BoW MLP | 67.65 | 41.69 | 64.51 | 4.93 |
| | BERT | 71.20 | 44.84 | 67.47 | 183.78 |
| | RoBERTa | 71.97 | 47.72 | 69.04 | 168.20 |
| | IntuitiveGraphLLM (RoBERTa) | 72.39 | 46.17 | 68.41 | 77.93 |
| | DeepSeek | 69.22 | 46.65 | 67.31 | 468.09 |
| | IntuitiveGraphLLM (DeepSeek) | 70.52 | 46.92 | 67.76 | 324.59 |
| | Llama | 69.09 | 44.37 | 66.31 | 225.44 |
| | IntuitiveGraphLLM (Llama) | **72.48** | 46.80 | 68.66 | 243.74 |
| BBC News | BoW MLP | 95.12 | 94.93 | 95.16 | 0.08 |
| | BERT | 96.74 | 96.57 | 96.75 | 1.37 |
| | RoBERTa | 96.74 | 96.62 | 96.76 | 0.97 |
| | IntuitiveGraphLLM (RoBERTa) | **99.19** | 99.10 | 99.19 | 2.98 |
| | DeepSeek | 98.37 | 98.42 | 98.37 | 6.89 |
| | IntuitiveGraphLLM (DeepSeek) | **99.19** | 99.24 | 99.19 | 19.19 |
| | Llama | 97.56 | 97.61 | 97.57 | 5.18 |
| | IntuitiveGraphLLM (Llama) | **99.19** | 99.18 | 99.19 | 2.07 |
| HND | BoW MLP | 78.46 | 72.12 | 78.05 | 0.05 |
| | BERT | 73.84 | 68.86 | 74.42 | 1.44 |
| | RoBERTa | 81.53 | 79.81 | 82.44 | 0.67 |
| | IntuitiveGraphLLM (RoBERTa) | **92.31** | 90.56 | 92.37 | 1.21 |
| | DeepSeek | 76.92 | 70.68 | 76.71 | 3.67 |
| | IntuitiveGraphLLM (DeepSeek) | 89.23 | 86.32 | 89.31 | 2.81 |
| | Llama | 80.00 | 76.84 | 80.66 | 2.81 |
| | IntuitiveGraphLLM (Llama) | **92.31** | 90.56 | 92.37 | 1.10 |
| IMDB | BoW MLP | 87.60 | 87.60 | 87.60 | 10.47 |
| | BERT | 88.80 | 88.79 | 88.80 | 22.61 |
| | RoBERTa | 90.28 | 90.26 | 90.28 | 18.13 |
| | IntuitiveGraphLLM (RoBERTa) | **95.72** | 95.71 | 95.72 | 38.97 |
| | DeepSeek | 93.60 | 93.60 | 93.60 | 130.71 |
| | IntuitiveGraphLLM (DeepSeek) | 94.40 | 94.40 | 94.40 | 363.25 |
| | Llama | 93.40 | 93.40 | 93.40 | 98.76 |
| | IntuitiveGraphLLM (Llama) | 94.52 | 94.52 | 94.52 | 40.74 |

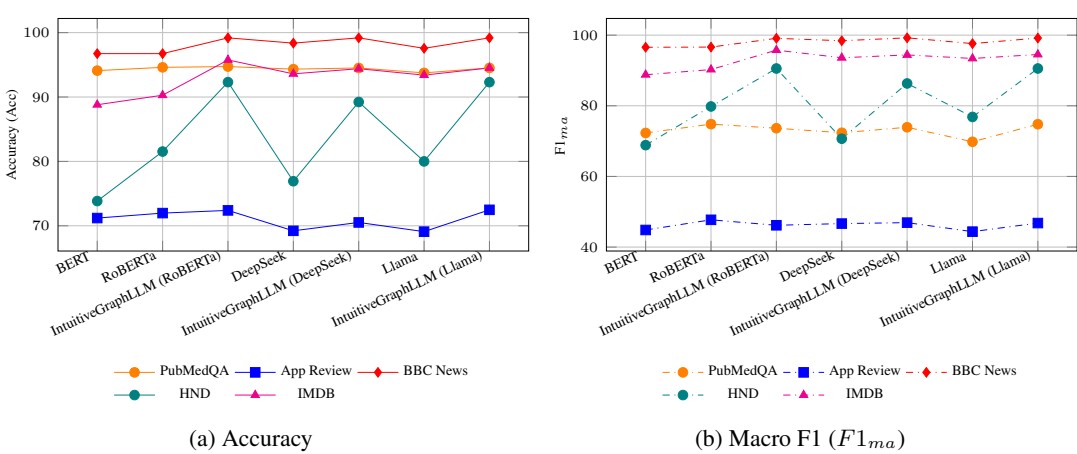

(a) Accuracy

(b) Macro F1 ($F1_{ma}$)

Figure 7: Comparison of overall accuracy and macro-F1 scores across five benchmark datasets, highlighting the performance of different models and their IntuitiveGraphLLM variants.

