# OpenReview forum: "IntuitiveGraphLLM: Intuitive Graph-based Text Representation with Large Language Model"
_ICLR.cc/2026/Conference — Submitted to ICLR 2026_

### Official Review · Reviewer_3ijq · 2025-10-24

**Soundness:** 2
**Presentation:** 2
**Contribution:** 2
**Rating:** 4
**Confidence:** 2

**Summary:**

This paper presents a framework called IntuitiveGraphLLM, which combines the power of Large Language Models with an innovative graph-based representation of text. The core idea behind IG-LLM is to construct Intuitive Graphs, which capture both structural context and semantic relevance between tokens, and then fuse these graphs with pretrained LLMs to enhance text representation and downstream performance.

**Strengths:**

1. The authors provide a clear and interpretable methodology for building the IGs, ensuring that the constructed graphs are both sparse and human-plausible. This interpretability is valuable for understanding how structural and semantic cues contribute to the performance of the model.
2. The paper reports thorough experiments across five benchmark datasets, covering a diverse range of tasks, including biomedical and commonsense reasoning. The approach consistently outperforms LLM-only models and graph-only models.

**Weaknesses:**

1. The framework introduces complexity by adding a parallel graph-based branch (IG construction, GAT encoding, and fusion). However, the performance benefit on the majority of datasets is negligible, calling the value of this added complexity into question. For example, on the PubMedQA dataset, the accuracy gain for the RoBERTa variant is only +0.12% and for the DeepSeek variant is +0.18%. On the App Review dataset, the gain for the RoBERTa variant is +0.42%. These minimal improvements are likely within the standard error range and do not present a strong case for adopting this complex hybrid model over a standard LLM.
2. While IntuitiveGraphLLM improves text representations by combining GATs with pretrained LLMs, this approach could introduce higher computational and memory overheads. Specifically, GATs require processing sparse graphs, and the computational complexity might grow significantly as the graph size increases, which can become a bottleneck for long texts or large datasets.

**Questions:**

See Weaknesses.

---

> ### Author Response · Authors · 2025-11-19
> **Response to concerns raised by 3ijq**
>
> Legend: Weakness (W), Response (R)
>
> ---
>
> $\textbf{R(W1:The framework introduces...complex hybrid model over a standard LLM.):}$ We appreciate the reviewer’s observation. The relatively small improvements on App Review and PubMedQA are expected and stem from a core characteristic of these datasets—their very short average token length (ATL) (please see Table 1). App Review and PubMedQA datasets have 27.60 and 17.44 ATL, respectively. In short documents, most contextual dependencies lie within a single sentence or local span, which modern LLM encoders already capture effectively. As a result, there is inherently limited room for additional relational reasoning, and the IntuitiveGraphLLM gains on these datasets remain small.
>
> In contrast, as shown in Table 3 and Figure 5, the contribution of IG becomes substantially larger for long-document datasets (e.g., HND, BBC, IMDB), where semantic and relational dependencies span distant tokens. In these settings, the semantic gating and relational aggregation in IntuitiveGraphLLM provide complementary global context that the LLM alone cannot capture. These results collectively confirm that while LLM embeddings dominate in short texts, IG plays a crucial role in long-document understanding, where structural reasoning and cross-sentence coherence are more important.
>
> Thus, IG is more effective for long-context reasoning—the setting where LLMs are most challenged. The empirical results robustly support this: IG contributions correlate strongly with document length, and the ablation results in Table 3 further confirm that IG is most impactful exactly in those scenarios where the LLM baseline struggles.
>
> ---
>
> $\textbf{R(W2: While IntuitiveGraphLLM improves text representations by...for long texts or large datasets):}$ We thank the reviewer for raising this important point. While IntuitiveGraphLLM introduces a graph-based branch, the framework was deliberately designed to keep the additional computational and memory overhead lightweight and bounded. Our analyses support this from several perspectives:
>
> (i) Semantic gating yields highly sparse graphs
>
> As reported in Section 4.2, the semantic gate removes 30–35% of candidate edges on average. This produces very sparse adjacency structures, so the GAT complexity scales as: $O(|E|.h)$ rather than as quadratic in the number of tokens. Because the gated IG graphs for long documents remain small graph expansion never becomes a bottleneck.
>
> (ii) The GAT branch is extremely lightweight relative to the LLM
>
> We computed the parameter distribution across all components (GAT, fusion layer, and RoBERTa). The results show: GAT parameters contribute only 0.19% of total trainable parameters, Fusion/FC layers add ~0.04%, LLM encoder dominates with >99% of total parameters. Thus, both computational and memory costs of adding IG are very small compared to the LLM forward/backward pass, which remains the dominant factor.
>
> (iii) The overhead is justified by large gains on long-document datasets
>
> Although IG introduces modest overhead, it yields substantial performance improvements where LLMs struggle most—long-document reasoning:
>
> -HND: +10.7% to +11.5% (Acc/F1)
>
> -BBC: +3–4%
>
> -IMDB: +1.5–2.3%
>
> In brief, the graph branch adds minimal computational cost (<1% of parameters), remains efficient due to semantic sparsity, and provides meaningful accuracy gains on long-document datasets. We will clarify these efficiency properties and provide the supporting parameter statistics in the revised version.

---

### Official Review · Reviewer_k3Xy · 2025-10-30

**Soundness:** 2
**Presentation:** 2
**Contribution:** 2
**Rating:** 4
**Confidence:** 3

**Summary:**

IntuitiveGraphLLM​ is a hybrid framework that enhances Large Language Models (LLMs) by integrating ​graph-based text representations​ that are both ​structurally grounded​ and ​semantically meaningful, while traditional text graphs (e.g., based on co-occurrence or sequential order) often include noisy or irrelevant edges, which can mislead reasoning.

**Strengths:**

1. The core idea—pruning a structural text graph with a semantic similarity gate—is an elegant and intuitive fusion of two well-established concepts: graph-based text representation and semantic similarity metrics.
2. The paper frames a critical limitation of LLMs—their difficulty with explicit relational reasoning—and proposes a hybrid solution that is distinct from more common approaches like retrieval-augmented generation (RAG) or chain-of-thought (CoT). It moves beyond simply using graphs as external knowledge bases and instead uses them to reconfigure the internal representation of the input text itself.
3. The structure from Introduction to Methodology, Experiments, and Conclusion is logical and easy to follow. The problem is well-motivated, and the solution is described in a modular fashion (Construction, Encoding, Fusion).

**Weaknesses:**

1.  The paper positions Intuitive Graphs (IGs) as a novel concept, but the core idea—pruning graph edges based on semantic similarity—is an established technique in graph-based NLP. The contribution lies in the specific application to LLM fusion, but this is not sufficiently distinguished from prior art.
2. The related work section (§2, Background) primarily contrasts IGs with simple structural graphs (BoW, sliding windows) but fails to engage with more sophisticated graph construction methods that also aim to reduce noise and incorporate semantics. This creates a strawman comparison.
3. Lack of Rigor in Semantic Gate Design and Analysis. The choice and analysis of the semantic gate, a central component, are surprisingly heuristic and underexplored. The threshold τis set to a fixed value of 0.3 across all datasets, described as a "conservative" choice to "avoid over-pruning." This is a major methodological shortcut. The paper acknowledges using Otsu's method and FDR to estimate dataset-specific thresholds (Appendix D) but does not utilize these estimates in the main experiments or analyze the sensitivity of performance to this critical hyperparameter. able 4 in Appendix D shows that the "optimal" threshold varies significantly by dataset and graph construction method (e.g., 0.4023 for BBC News with Window5, up to 1.00 for PubMedQA with Window3). Ignoring this variation suggests the reported gains might not be optimal and that the method is sensitive to an un-tuned parameter.
4. The ablation study (Section 5.3) is limited and fails to isolate the impact of the most important component: the ​semantic gate​ itself.

**Questions:**

See above.

---

> ### Author Response · Authors · 2025-11-19
> **Response to concerns raised by k3Xy**
>
> Legend: Weakness (W), Response (R)
>
> $\textbf{R(W1:The paper positions Intuitive Graphs..distinguished from prior art):}$ We thank the reviewer for this observation. While pruning graph edges based on semantic similarity is indeed known in graph-based NLP, the proposed IntuitiveGraphLLM introduces a $\textbf{novel and methodologically distinct}$ use of this idea within an LLM–graph hybrid architecture. The contribution of IGs is not the cosine similarity itself, but the way semantic gating is operationalized and coupled with LLMs. The IG is defined by three design components that collectively distinguish it from prior work: (1) Structure-aware initialization, (2) Semantic gating as a graph topology-altering mechanism, and (3) LLM–Graph fusion for complementary reasoning.
>
> To further validate that the IGs provide meaningful signals, we permuted the IG features. We re-trained the IntuitiveGraphLLM (RoBERTa) model using these permuted features. As shown in Table X1, the A and F1 scores dropped by approximately 11% on HND, 0.8% on IMDB, and 2.4% on BBC. These declines suggest that aligned IG features carry non-trivial semantic signals that aid contextual understanding beyond text embeddings alone.
>
> Table X1: Performance comparison between the original and the permuted IG features
>
>              | Original   | Permuted IG | Impact
> Dataset |  A     | F1(ma) |   A   |   F1(ma)    | ∆A  | ∆F1(ma)
>
> BBC     | 99.19 | 99.10  | 96.75 |	96.84 | -2.44  | -2.26
>
> App      | 72.39 | 46.17  | 72.09 |  47.09 | -0.30  | +0.92
>
> HND      | 92.31 | 90.56  | 81.54 |	78.90 | -10.77 | -11.66
>
> IMDB    | 95.76 | 95.76  | 94.96 |	94.96 | -0.80  | -0.80
>
> PubM. | 94.73 | 73.65  | 94.52 |  74.12 | -0.21  | +0.47
>
> ---
>
> $\textbf{R(W2: The related work section (§2, Background)...creates a strawman comparison):}$ We agree that the related work section should engage more deeply with sophisticated graph construction and graph-enhanced language modeling methods. We will revise Section §2 to include an expanded discussion of recent representative approaches, highlighting their objectives, assumptions, and design differences relative to our method.
>
> -We conducted additional experiments using both (i) an LLM embedding model and (ii) Graphormer, a graph model, under the same IG settings to compare our results.
>
> Table X2. Performance of IG with Jina Embeddings [1]
>
> Dataset   |  Acc  | F1(wg)  | F1(ma)
>
> HND| 64.62 | 58.68 | 43.05
>
> App  | 68.31 | 63.65 | 41.16
>
> BBC | 34.96 | 30.97 | 30.85
>
> PubM. | 93.04 | 90.25 | 53.38
>
> IMDB| 89.88 | 89.88 | 89.87
>
> -These results are significantly inferior to (i) our IntuitiveGraphLLM performance and (ii) the baseline LLM backbones. In particular, performance drops sharply on long-document datasets (e.g., HND, BBC), where global relational reasoning is most crucial.
>
> -Similar to the embedding model, Graphormer [2] fails to outperform IntuitiveGraphLLM on any dataset
>
> Table X3. Performance of Graphormer Models
>
> Dataset|  Acc  | F1(ma)
>
> HND | 72.31 | 41.96
>
> App | 68.04 | 34.79
>
> BBC | 21.95 |18.17
>
> PubM. | 92.86 |48.15
>
> IMDB | 51.92 | 34.18
>
> -These results reinforce that IntuitiveGraphLLM is not simply a graph encoder, but an effective model that leverages both token-level semantics and graph-level relational knowledge.
>
> [1] Jina Embeddings: A Novel Set of High-Performance Sentence Embedding Models
>
> [2] Do transformers really perform bad for graph representation?
>
> ---
>
> $\textbf{R(W3:Lack of Rigor in Semantic Gate ...untuned parameter):}$ In light of your comment, we conducted additional experiments with the thresholds suggested by the OTSU and FDR approaches.
>
> Table X4. Performance IntuitiveGraphLLM (RoBERTa) using OTSU/FDR-Derived τ Values
>
> Dataset	| τ |	A  |	F1(wg) | F1(ma)
>
> HND | 0.42 | 92.31 | 92.24 | 90.23
>
> IMDB | 0.41 | 95.76 | 95.76 | 95.75
>
> BBC | 0.42 | 98.37 |	98.39 | 98.22
>
> App | 0.52 | 72.39 |	67.83 | 45.20
>
> PubM. | 0.48 | 94.37 | 93.65 | 73.65
>
> -These results are nearly identical to those reported in the main submission using τ = 0.3, and in some cases, the difference is within 0.1–0.3%. This demonstrates that the semantic gate is robust to a wide range of τ values. Our choice of τ = 0.3 was intentionally conservative to avoid over-pruning, but the additional results confirm that performance is not sensitive to moderate deviations in τ.
>
> ---
>
> $\textbf{R(W4: The ablation study (Section 5.3) is limited...gate itself)}$: In the paper, our ablation study focused on removing the entire IG module to demonstrate its contribution within the overall IntuitiveGraphLLM framework.
>
> -Additional component-level analyses, including comparisons of IG construction methods, node embedding choices, and different GAT depths across datasets, have been presented in the Appendix.
>
> -In addition to the reported ablations, we compared the model performance between the permuted IG features and the original. To demonstrate that the IG features are responsible for preserving meaningful relational information.

---

### Official Review · Reviewer_zVku · 2025-10-31

**Soundness:** 2
**Presentation:** 3
**Contribution:** 2
**Rating:** 2
**Confidence:** 3

**Summary:**

This paper proposes IntuitiveGraphLLM that incorporates Intuitive graphs to encode graph by reducing noise from over-connectING unrelated tokens. Comprehensive experiments further verify the effectiveness of the proposed framework.

**Strengths:**

+ This paper is well-organized and easy to follow
+ This paper conducts comprehensive experiments to verify the effectiveness of IntuitiveGraphLLM

**Weaknesses:**

- Lack a comprehensive comparison with current State-of-Art graph embedding models.(e.g. ENGINE[1], Unigraph[2], TAPE[3])
- Lack an analysis on different graph encoder types.
- Limited task scope. The paper only evaluates IntuitiveGraphLLM on node-level tasks, omitting an evaluation on edge-level tasks, such as link prediction.


[1]Efficient Tuning and Inference for Large Language Models on Textual Graphs

[2]UniGraph: Learning a Unified Cross-Domain Foundation Model for Text-Attributed Graphs

[3]Harnessing Explanations: LLM-to-LM Interpreter for Enhanced Text-Attributed Graph Representation Learning

**Questions:**

- For some datasets, the improvement seems minor (less than 1%); please provide the variance across multiple runs.
- For the LLM backbones, they are not specially designed for encoding text sequences, I am curious whether the LLM-based embedding models (e.g., NV-EmbEd-V2, Jina-Embeddings) will further improve the performance.
- Compare IntuitiveGraphLLM with other graph embedding models.
- As the paper states, their selection stresses domain shift, which makes it important to examine the model’s performance in a transfer setting (e.g., from PubMedQA to IMDB).

---

> ### Author Response · Authors · 2025-11-18
> **Response to concerns raised by zVku**
>
> Legend: Weakness (W), Question (Q), Response (R)
>
> R(W: Lack a comprehensive comparison...): Thank you for pointing out these important points. We will discuss about the recent graph encoder models in our paper. We have already conducted some experiments with encoder models, graph encoders and several random seeds discussed in answering your questions.
>
> -Edge-level tasks such as link prediction or edge classification require fundamentally different supervision signals and graph structures. However, the IntuitiveGraphLLM concept is not limited to node-level tasks. Because IG produces explicit semantic edges and graph embeddings, the framework could be naturally extended to edge prediction by defining learning objectives over gated edges. We acknowledge this as an interesting direction for future research.
>
> ---
>
> R(Q1:For some datasets...multiple runs): For the datasets (App Rev. and PubMedQA) where IntuitiveGraphLLM obtained minor gains, we performed additional experiments using five random seeds (42, 56, 78, 100, 123). As shown in Table X1, the performance remains stable across runs, variance is extremely small, confirming that the observed results are not due to random initialization or training noise.
>
> TableX1. Mean ± Standard Deviation across Five Seeds
>
> Dataset              |           Acc         |       F1(wg)     |     F1(ma)
>
> PubMedQA        |	94.59 ± 0.12 |  93.80 ± 0.06 |	73.69 ± 0.59
>
> App Rev.            |	72.19 ± 0.05 |	68.67 ± 0.24 |	46.72 ± 1.00
>
> The results show that the IntuitiveGraphLLM performance is consistent, even though the gains on these datasets are small. The reason for the limited improvement is the short text length in App Rev. and PubMedQA (Table 1). In short text, the LLM encoder already captures most of the local contextual dependencies, leaving little room for IG-based relational reasoning to add value. In contrast, IntuitiveGraphLLM yields substantial improvements on long-document datasets.
>
> ---
>
> R(Q2: For the LLM backbones...performance): We conducted additional experiments using Jina-Embeddings under the same IG construction and fusion settings. The results across datasets are shown below:
>
> Table X2. Performance of IG with Jina Embeddings
>
> Dataset   |  Acc  | F1(wg)  | F1(ma)
>
> HND	  | 64.62 | 58.68 | 43.05
>
> IMDB	  | 89.88 | 89.88 | 89.87
>
> App Rev.  | 68.31 | 63.65 | 41.16
>
> PubMedQA  | 93.04 | 90.25 | 53.38
>
> BBC | 34.96 | 30.97 | 30.85
>
> -These results are significantly inferior to (i) our IntuitiveGraphLLM performance and (ii) the baseline LLM backbones. In particular, performance drops sharply on long-document datasets (e.g., HND, BBC), where global relational reasoning is most crucial.
>
> -Models like RoBERTa, and Llama provide rich token-level contextual embeddings that synergize effectively with IG. In contrast, embedding models [1,2] collapse the entire text into a single sentence-level vector, discarding the token-level information required for IG construction, semantic gating, and message passing. As a result, the IG and embedding branches become semantically misaligned, leading to poor fusion and significantly weaker results.
>
> [1] NV-Embed: Improved Techniques for Training LLMs as Generalist Embedding Models
>
> [2] Jina Embeddings: A Novel Set of High-Performance Sentence Embedding Models
>
> ---
>
> R(Q3: Compare IntuitiveGra ...embedding model): In response to the reviewer’s suggestion, we evaluated a stronger graph representation model, Graphormer [3], under the same settings. The results are shown below.
>
> Table X3. Performance of Graph Embedding Models
>
> 	       Graphormer      GAT
> Dataset    |  Acc  | F1ma  |  Acc  | F1ma
>
> HND	   | 72.31 | 41.96 | 72.31 | 41.96
>
> IMDB	   | 51.92 | 34.18 | 77.28 | 77.27
>
> App Rev.    | 68.04 | 34.79 | 68.23 | 33.72
>
> PubMedQA  | 92.86 | 48.15 | 94.34 | 66.88
>
> BBC | 21.95 | 18.17 | 56.10 | 48.10
>
> Across datasets, neither Graphormer nor GAT alone comes close to the performance of any IntuitiveGraphLLM variant. These findings highlight two key points: (i) Graph-only models are insufficient for text understanding, (ii) The strength of IntuitiveGraphLLM lies in the fusion of LLM embeddings with IG relational structure. These results reinforce that IntuitiveGraphLLM is not simply a graph encoder, but an effective model that leverages both token-level semantics and graph-level relational knowledge.
>
> [3] Do transformers really perform bad for graph representation?
>
> ---
>
> R(Q4: As the paper states, their...setting): Our goal in this work was to design a scalable and generalizable hybrid architecture that can integrate graph-based relational reasoning with a wide variety of LLM backbones.
>
> -We agree that cross-dataset transfer (e.g., training on PubMedQA and testing on IMDB) would be a valuable perspective. However, such cross-domain transfer introduces confounding factors: task mismatch, label-space differences, and inherent domain divergence unrelated to the IG architecture. In the future, we will explore the transfer settings with IntuitiveGraphLLM.

---

### Official Review · Reviewer_9kmx · 2025-11-01

**Soundness:** 2
**Presentation:** 2
**Contribution:** 2
**Rating:** 2
**Confidence:** 4

**Summary:**

This paper proposes a new hybrid framework that fuses graph-based relational reasoning with LLM-based contextual understanding to improve semantic and logical comprehension in text-based tasks.The key idea is to prune structure-based text graphs (like windowed or sequential graphs) using a semantic gating mechanism based on cosine similarity between token embeddings, yielding sparser and more semantically meaningful graphs.

**Strengths:**

- Comprehensive experiments.
- The proposed semantic gating is interpretable.

**Weaknesses:**

- Ambiguous conceptual novelty.
- The conclusions are not well supported by the experimental evidence.

**Questions:**

1. The semantic gating is conceptually straightforward (cosine similarity filtering), not novel. How does the proposed semantic gate differ fundamentally from prior semantic reweighting or graph pruning mechanisms?
2. The fusion is a simple concatenation of [GAT output; LLM output]. This is a simple soft prompting strategy. Have the authors tested whether improvements persist if IG features are randomly shuffled before fusion?
3. The authors mix encoder-only (RoBERTa) and decoder-only (Llama, DeepSeek) models under one framework. A justification for this design choice would be appreciated.
4. In Table 3, removing the LLM leads to huge drops, but removing IG causes smaller drops — sometimes even an increase in F1. For example, in PubMedQA, w/o IG gives higher F1 (74.81 vs. 73.65). This undermines the claim that IGs are always the key driver of performance. A discussion for these observations would be appreciated.
5. The term Intuitive Graph is rhetorically strong but methodologically vague.

---

> ### Author Response · Authors · 2025-11-18
> **Response to concerns raised by 9kmx**
>
> Legend: Question(Q), Response(R)
>
> R(Q1: The semantic gating is...pruning mechanisms?): The novelty of our approach does not lie in the use of cosine similarity (CS) itself, but in where and how it is applied within the IntuitiveGraphLLM framework. Specifically, our semantic gate is principled and statistically estimated, rather than a hand-picked. It actively modifies the graph topology before information propagation and remains enforced throughout the attention process, resulting in measurable sparsity, reduced noise, and consistent downstream gains across different graph constructions and LLM backbones. It ensures that relevant tokens, both semantically and structurally, are propagated.
>
> The CS-based gating differs fundamentally from prior semantic reweighting/pruning methods that (a) retain all edges for attention computation, (b) rely on heuristic or fixed cutoffs, or (c) entangle semantic information directly into attention scores rather than into the adjacency structure itself. Moreover, traditional reweighting mechanisms merely adjust edge strengths while keeping all connections intact, which often results in dense graphs with noisy or redundant relations.
>
> ---
>
> R(Q2:The fusion is a simple...before fusion?): We performed control experiments (across datasets) where the Intuitive Graph (IG) embeddings were randomly permuted before fusion with the LLM outputs. Table X1 reports the correlation metrics between the original and permuted IG embeddings. The near-zero values of cosine similarity (CS), mean Pearson (MP), and mean Spearman (MS) correlations confirm that the permutation completely disrupts the semantic structure.
>
> Table X1:Correlation metrics before vs. after IG feature shuffling
>
> Dataset | CS | MP |  MS
>
> HND | 0.0013 | 0.0012 | 0.0023
>
> IMDB | 0.0161 | 0.0133 | 0.0106
>
> BBC | 0.0203 | 0.0146 | 0.0116
>
> App Rev. | 0.0128| 0.0112 | 0.0076
>
> PubMedQA | 0.005 | -0.0091 | 0.0006
>
> We re-trained the IntuitiveGraphLLM (RoBERTa) model using these permuted features. As shown in Table X2, the A and F1 scores dropped by approximately 11% on HND, 0.8% on IMDB, and 2.4% on BBC. These declines indicate that aligned IG features carry non-trivial semantic signals that aid contextual understanding beyond text embeddings alone.
>
> Table X2: Performance comparison between original and permuted IG features
>
>            | Original   | Permuted IG  | Impact
> Dataset | A     | F1(ma) |   A   |   F1(ma)    | ∆A  | ∆F1(ma)
>
> BBC     | 99.19 | 99.10  | 96.75 |	96.84 | -2.44  | -2.26
>
> App      | 72.39 | 46.17  | 72.09 |  47.09 | -0.30  | +0.92
>
> HND      | 92.31 | 90.56  | 81.54 |	78.90 | -10.77 | -11.66
>
> IMDB    | 95.76 | 95.76  | 94.96 |	94.96 | -0.80  | -0.80
>
> PubM.|94.73 | 73.65  | 94.52 |  74.12 | -0.21  | +0.47
>
> For App Rev. and PubMedQA, the performance remained similar, likely because their ATL is short (Table 1 in the paper), limiting the graph’s ability to encode additional structure. Consistent with our ablation results (Table 3 in the paper), IG contributes more strongly to long-document datasets, confirming that IntuitiveGraphLLM is especially effective when semantic and relational dependencies extend across distant tokens.
>
> ---
>
> R(Q3: The authors mix encoder-only...would be appreciated.): Our goal was to assess the scalability and generalizability of the proposed IntuitiveGraphLLM across different LLM architectures. To this end, we deliberately included both encoder-only models and decoder-only models in our experiments. This design choice demonstrates that the proposed framework, based on semantic gating and hybrid representation, is model-agnostic and can integrate effectively with distinct LLM structures. Showing consistent improvements across both categories validates that IntuitiveGraphLLM does not rely on any architecture-specific property and is therefore scalable, transferable, and adaptable to a wide range of pretrained LLMs.
>
> ---
>
> R(Q4: In Table 3, removing the LLM...be appreciated):  As noted in our response to Question #2, the relatively small impact of removing IG features in App Review and PubMedQA stems from the short ATL of these datasets. When documents are short, the contextual relationships among tokens are already captured effectively by the LLM encoder, leaving limited scope for additional graph-level reasoning. Consequently, removing the IG component or shuffling IG features produces negligible changes or minor fluctuations in F1 and A scores.
>
> ---
>
> R(Q5: The term Intuitive Graph...methodologically vague.): The term Intuitive Graph was chosen not for rhetorical emphasis, but to capture the core methodological intuition of our framework, constructing graph structures that align with the intuitive semantic relationships present in text rather than relying purely on rigid structural heuristics. Empirically, the proposed IG leads to measurable performance gains on long-document datasets (e.g., HND), confirming that the IG concept is both methodologically sound and empirically validated.

---

### Meta-Review · Area_Chair_4xrm · 2025-12-09

**Summary:**

This paper proposes the framework IntuitiveGraphLLM to improve the performance of LLM by combining them with intuitive graphs. These graphs are constructed by pruning structure-induced edges using semantic gates based on cosine similarity between token (or span) embeddings. The aim is to remove irrelevant edges while preserving connections with structural and semantic relevance.

While the idea of fusing graph-based relational reasoning with contextual understanding based on LLMs is conceptually appealing, all reviewers raised serious concerns about the lack of genuine innovation (cosine similarity-based graph edge pruning is considered a standard, well-established technique in NLP), weak experimental baselines (lack of comparison with state-of-the-art text attribute graph models), and limited gains. The authors' rebuttals failed to address these core technical and experimental shortcomings, often offering vague responses or promises of future updates rather than specific data. Therefore, the paper requires significant revisions.

**Reviewer Concerns:**

### Addressed Concerns:

- Reviewer **9kmx** asked if gains were merely due to soft prompting rather than graph structure. The authors conducted a control experiment shuffling the IG features, which resulted in significant performance drops (e.g., -10.77% on HND). This effectively demonstrated that the aligned graph features carry meaningful signal.
- Reviewer **zVku** requested variance statistics for datasets with minor gains. The authors provided mean/standard deviation results across five seeds, demonstrating training stability.
- Reviewer **k3Xy** criticized the fixed semantic threshold ($\tau=0.3$) as lacking rigor. The authors performed sensitivity analyses using Otsu’s method and FDR to estimate thresholds. The results showed that performance was robust to threshold variations, validating their conservative choice of 0.3.

### Outstanding Concerns

- Reviewer **zVku** explicitly requested comparisons with ENGINE and UniGraph. The authors only vaguely indicate that recent graph encoder models will be discussed in the paper. The failure to benchmark against the specific models cited by the reviewer leaves the claim of superiority over graph-based methods unsupported.
- Reviewers **9kmx** & **k3Xy** argued that semantic gating via cosine similarity is a straightforward, established technique. The authors argued that the application within this specific hybrid framework is novel. However, this does not fully resolve the concern that the "Intuitive Graph" is essentially a thresholded similarity graph.
- Reviewers **9kmx** & **3ijq** noted the lack of improvement on PubMedQA/App Reviews. The authors explained this is due to LLMs already handling local context well. While the explanation is sound, it confirms that the proposed method is not a general-purpose improvement, limiting its broad utility.

**Reviewer Scores:**

- **Reviewer 9kmx (Current: 2):** **Prediction: 2.** The reviewer would likely appreciate the feature shuffling experiment which ruled out the "soft prompting" hypothesis. However, the core objection regarding "ambiguous conceptual novelty"  remains unaddressed. The reviewer would likely still view the method as too incremental for acceptance.
- **Reviewer zVku (Current: 2):** **Prediction: 2.** This reviewer specifically asked for comparisons to **ENGINE** and **UniGraph**. The authors did not provide these comparisons. This failure to address the direct request regarding SOTA baselines would likely result in the score remaining unchanged.
- **Reviewer k3Xy (Current: 4):** **Prediction: 4.** The sensitivity analysis regarding the threshold  directly addressed this reviewer's complaint about rigor. However, the "strawman comparison" concern regarding baselines and “ablation study” concern would likely prevent a higher score.
- **Reviewer 3ijq (Current: 4):** **Prediction: 4.** The authors provided a strong defense regarding computational overhead (showing it is negligible) and justified the complexity via the gains on long-document tasks. This addresses the reviewer's main hesitation, but the marginal gains on the experimental datasets remain a dampener.

---

### Decision · Program_Chairs · 2026-01-26

Reject